# Klotho and Mesenchymal Stem Cells: A Review on Cell and Gene Therapy for Chronic Kidney Disease and Acute Kidney Disease

**DOI:** 10.3390/pharmaceutics14010011

**Published:** 2021-12-21

**Authors:** Marcella Liciani Franco, Stephany Beyerstedt, Érika Bevilaqua Rangel

**Affiliations:** 1Albert Einstein Research and Education Institute, Hospital Israelita Albert Einstein, Sao Paulo 05652-900, Brazil; marcella.franco07@gmail.com (M.L.F.); stephany.beyerstedt@gmail.com (S.B.); 2Nephrology Division, Federal University of São Paulo, Sao Paulo 04038-901, Brazil

**Keywords:** chronic kidney disease, acute kidney injury, Klotho, mesenchymal stem cells

## Abstract

Chronic kidney disease (CKD) and acute kidney injury (AKI) are public health problems, and their prevalence rates have increased with the aging of the population. They are associated with the presence of comorbidities, in particular diabetes mellitus and hypertension, resulting in a high financial burden for the health system. Studies have indicated Klotho as a promising therapeutic approach for these conditions. Klotho reduces inflammation, oxidative stress and fibrosis and counter-regulates the renin-angiotensin-aldosterone system. In CKD and AKI, Klotho expression is downregulated from early stages and correlates with disease progression. Therefore, the restoration of its levels, through exogenous or endogenous pathways, has renoprotective effects. An important strategy for administering Klotho is through mesenchymal stem cells (MSCs). In summary, this review comprises in vitro and in vivo studies on the therapeutic potential of Klotho for the treatment of CKD and AKI through the administration of MSCs.

## 1. Introduction

The Klotho gene was first introduced and described in 1997 by Kuro, et al., as an anti-aging gene. They reported a mutant Klotho-deficient animal model, which presented phenotypes similar to age-related events in humans, such as reduced lifespan, alongside vascular calcification and cardiovascular disease [1,2]. Named after the Greek goddess of fate in mythology [1], Klotho is a 140 kDa protein and it has a high homology with β-glucosidases [1]. This review comprises information about αKlotho—one of this protein family’s isoforms—referred to in the present study as “Klotho”.

In humans, Klotho is present in two distinct isoforms—anchored on the membrane protein or as a soluble protein. Membrane-anchored Klotho is a single-pass transmembrane protein, of which the large extracellular domain is composed of two repeated sequences of 440 amino acids, named K11 and K12. These sequences can be proteolytically cleaved by different enzymes, leading to cleaved Klotho—one of the soluble Klotho forms [1,3,4,5,6,7]. The soluble form of this protein may be also produced from alternative splicing from its gene, leading to the formation of secreted Klotho, although this form of Klotho has not been detected in vivo to date [8,9,10] and studies indicate that it might not actually be secreted, because it is molecularly degraded [10].

In adult humans, the membrane-anchored form of Klotho is expressed especially in the parathyroid glands, sinoatrial node, choroid plexus and, mainly, in distal tubules in the kidneys [11]. Soluble Klotho, on the other hand, is present in systemic circulation and it has pleiotropic roles, acting as an endocrine factor with both renal and extrarenal effects [12]. It can be detected in the blood, urine and cerebrospinal fluid [9,13,14,15]. Both isoforms are of pivotal importance for homeostasis, as will be further addressed. In short, they impact on the balance of phosphate and other ions [16,17,18], for example, through the regulation of the absorption of different molecules such as calcium [8,19], as well as ion channels such as transient receptor potential V5 (TRPV5), sodium-phosphate cotransporter (NaPi2a) and, indirectly, sodium-chloride cotransporter (NCC) [20,21,22]. Klotho is also important for the cardiovascular system [23] and it participates in other biological events [24], as it will be discussed next.

Concerning its importance in embryo development, some data with animal models have indicated that Klotho is expressed since the early stages of life. Mangos, S., et al., for instance, have reported that this protein is detected in zebrafish within 24 h postfertilization (hpf) in the brain and in the pronephric ducts, which are the primitive tubules [25]. Likewise, similar findings were obtained in rodent models [26,27]. In adult zebrafish, it was observed that the expression of Klotho is maintained in the mesonephric kidney [25].

The levels of Klotho decrease with aging [28] and in the case of kidney diseases, such as chronic kidney disease (CKD) [29] and acute kidney injury (AKI). This decline is associated, for example, with the loss of renal mass and 1,25(OH)2D synthesis [3]—which, in physiology, enhances Klotho expression in the kidneys, as albuminuria, angiotensin II and proinflammatory molecules lead to the reduction of its expression [30,31].

Klotho is, in this way, an essential factor to be investigated in both the physiology and pathology of renal diseases.

## 2. Klotho and Chronic Kidney Disease

### 2.1. Chronic Kidney Disease

Chronic Kidney Disease (CKD) is a multifactorial disease [7], defined by the Kidney Disease Improving Global Outcomes Work Group (KDIGO) in 2012 as the presence of either a reduction in kidney function and/or albuminuria, that is, the abnormal excretion of albumin in the urine, for at least three months [32,33]. This illness is considered a public health issue, with prevalence rates of about 8–16% worldwide [34], and this percentage continues to rise, especially due to the aging population and the increasing incidence of type-2 diabetes, for instance [2]. Despite its several etiologies, some of the risk factors for CKD are diabetes, a family history of CKD, heart disease, high blood pressure and obesity. As early CKD might not cause any symptoms, measurement of both the serum creatinine level and protein in the urine are important methods for CKD diagnosis [35]. A key aspect of CKD is the progressive deterioration of renal function, which often leads to end-stage kidney disease (ESKD) [36], resulting in renal failure—mainly treated with either dialysis or renal transplant—and even to extrarenal complications, such as cardiovascular disease (CVD) [35]. Hence, this disorder is associated with morbidity and mortality; furthermore, it represents a high socioeconomic concern for health systems [33].

Although there are tests to detect CKD, as mentioned above, there is a scarcity of biomarkers to diagnose this disease early and precisely and avoid its progression to ESKD and other complications [37,38]. Thus, some potential early biomarkers for CKD have been studied over the years, as reviewed by Shabaka, A., et al. [37], such as Dickkopf-3 (DKK-3), a glycoprotein associated to the degree of tubulointerstitial fibrosis, of which high levels in the urine indicate an elevated risk for a reduction in eGFR within a year [39]. Neutrophil gelatinase-associated protein (NGAL) [40] is another example of a potential biomarker for CKD. These and other possible biomarkers, however, do not represent increased advantages in CKD diagnosis when compared to the traditional markers analyzed. Therefore, the study of new molecules for diagnostic purposes is still necessary [37].

Regarding therapeutic approaches, there are non-pharmacological options for the treatment of CKD, such as weight reduction and blood pressure control. Still considering the study of Shabaka, A., et al. [37], though, CKD is a complex disease, so the use of some drugs alongside non-pharmacological treatments is also important. Some examples of pharmacological treatment that can be used are sodium-glucose-cotransporter 2 inhibitors, such as empagliflozin—which induce both renal and extrarenal benefits for patients [37]—and renin-angiotensin-aldosterone system (RAAS) inhibitors, such as blockers for angiotensin II receptors and inhibitors for angiotensin-converting enzymes [37]. These are relevant therapeutic options, due to the fact that they are able to reduce the loss of eGFR, through the decline of intraglomerular pressure. Moreover, new drugs are under development, such as a mineralocorticoid receptor antagonist and potassium-lowering therapies [37]. Considering the complexity of CKD, however, the development of new therapeutic strategies is of pivotal importance.

Some of the main characteristics of CKD are chronic inflammation, hypoxia and oxidative stress, which contribute to structural and functional changes in the kidneys, resulting in glomerular, tubular and vascular injuries [33]. Thus, this disease is responsible for disturbances in mineral metabolism [41], with hyperphosphatemia and mineral-bone disorders [2] being some of its consequences.

During the progression of CKD, proinflammatory factors—interleukin 6 (IL-6) and tumor necrosis factor (TNF) [7], for example—show increased levels in the kidneys. Another significant proinflammatory molecule increased in CKD is the nuclear factor κB (NF-κB), a transcription factor related to the upregulation of cytokine expression [1]. There is also an activation of macrophages, alongside T-cell recruitment. As a consequence, these cell types and tubular epithelial cells produce profibrotic molecules. As such, transforming growth factor β (TGF-β) is one of the most influential mediators in the fibrosis process in CKD, since it stimulates the accumulation of matrix proteins and the epithelial-to-mesenchymal transition (EMT), inhibits matrix degradation and regulates myofibroblast activation [42,43,44,45]. In this context, injured tubular epithelial cells undergo a dedifferentiation process and lose their transport function and polarity. Furthermore, they synthesize the extracellular matrix. The final result of this microenvironment is the development of renal fibrosis [7].

In spite of the above description, the mechanisms relating to CKD have not yet been fully elucidated. Strong evidence, however, has pointed out the involvement of Klotho in this process, as will be addressed next.

### 2.2. Klotho in Chronic Kidney Disease

It has been observed in several studies that there is a decrease in Klotho levels (mRNA and protein) [2] both in animal models and individuals with CKD and renal failure [7]. The sustained Klotho imbalance in its soluble and membrane-anchored forms is associated with a decline in renal function, even in early CKD, when urinary excretion of Klotho is present in patients with this disease [33,38]. Klotho expression and levels become lower during CKD progression [38], as the estimated glomerular filtration rate (eGFR) decreases [34]. These changes in eGFR, as part of the natural history of CKD, are reflected by soluble Klotho and because of this, among other reasons, this protein could be used as an indicator for the evolution of CKD [33], for the degree of renal insufficiency in general and even for extrarenal complications [36]. Some evidence implies that the depletion of Klotho in murine models is positively correlated with persistent and increased inflammation [7]. Furthermore, it has been demonstrated that CKD is also associated with a decrease in Klotho expression [2]. Although the consequences of Klotho deficiency are not fully understood yet, it has been evidenced that renal and vascular cells senescence are some of the outcomes of this situation [2]. Likewise, a reduction in Klotho levels is associated with CKD inflammation and increased albumin excretion in patients, alongside a higher risk for some extrarenal complications, such as CV diseases and mortality. Interestingly, the restoration of Klotho levels in rodent animals through the administration of soluble Klotho, or the activation of endogenous protein, for instance, promotes the reduction of renal fibrosis, EMT and a decrease in oxidative stress and the inflammatory burden [2]. In conjunction with these data, further analyses identified that the overexpression of Klotho can lead to the enhancement of phosphaturia and to a decrease in vascular calcification in vivo, as well as to an improvement in renal function [36].

In regard to this topic, studies conducted with rodents have strongly suggested that the administration of soluble Klotho is a safe approach [19,46], although the complete spectrum of effects promoted by Klotho is still being evaluated. Likewise, in a rodent model of glomerulonephritis, which overexpresses exogenous Klotho, there is evidence of the improvement of proteinuria and serum creatinine levels. Moreover, there is also evidence of a reduction in renal cellular senescence—through a decline in β-galactosidase activity—as well as a restoration of mitochondrial activity in the cortex and the attenuation of mitochondrial damage, through cytochrome C enzyme activity reestablishment and a reduction in mitochondrial DNA damage, respectively [47]. The same study also reported a decrease in both oxidative stress and apoptosis in renal tissue. It is important to mention that viral gene delivery of Klotho, on the other hand, has not been proven to be safe in clinical studies yet [48], although it has been shown to be effective in preclinical studies, as will be further addressed in this review. Taken together, these results suggest that Klotho is a sensitive biomarker for CKD and renal function in general due to the fact that this protein level is reduced since the early stages of CKD and accompanies the decrease in eGFR. In addition, the reduction in Klotho is associated, as shown by different studies and as discussed above, with some of the characteristics of CKD, such as cellular senescence, albuminuria and cardiovascular disease.

Thus, Klotho deficiency is not only a biomarker for CKD, but also a pathogenic factor in the development, progression and complication of this disorder [38]. Importantly, preclinical data have strongly suggested that the increase in Klotho levels is safe and can mitigate fibrosis, vascular calcification, proteinuria, creatinine levels and oxidative stress, among other biological responses that are unbalanced in CKD. Therefore, further clinical studies are still necessary in order to shed light on Klotho efficiency and safety on CKD treatment, but current data strongly point to this molecule as a potential therapeutic target and its restoration levels as an approach for the treatment of CKD. Although the exact mechanisms through which Klotho influences CKD have not yet been well elucidated, several studies have addressed this issue, as discussed below.

#### 2.2.1. Klotho and FGF-23

Anchored on the membrane form of Klotho is a co-receptor for fibroblast growth-factor 23 (FGF-23), a hormone that is produced by cells residing in bone, namely osteocytes, to target a distant organ, the kidney. The Klotho/FGF-23 complex is responsible for, among other biological responses, the activation of extracellular signal-regulated kinase (ERK)1/2 and serum/glucocorticoid-regulated kinase (SGK)1. These signaling pathways downregulate the expression of the main sodium phosphate cotransporter in proximal tubules, NaPi-2a, on the membrane of tubular cells, resulting in phosphaturia, that is, renal phosphate excretion in urine [7]. Experiments with mice suggest that the phosphaturic effect promoted by FGF-23 depends on Klotho, although the molecular mechanisms of this regulation are not yet fully understood [22,49,50]. FGF-23 also reduces the levels of 1,25-dihydroxyvitamin D3 (1,25-(OH)2VD3), leading to decreased intestinal reabsorption of phosphate [38]. Hence, the Klotho/FGF-23 axis is important for the ion balance and homeostasis [36]. A study conducted on 152 patients with CKD has suggested that reduced Klotho levels aggravate phosphaturia [33]. Furthermore, according to the literature, imbalance in the FGF-23-Klotho pathway and its consequent hyperphosphatemia is connected to the progression of CKD [2]. At the same time, some investigations indicate that a deficiency of Klotho limits the regulation of FGF-23 [7].

Concerning other minerals, some studies propose that, in distal tubules, Klotho/FGF-23 complexes are responsible for the modulation of calcium and sodium reabsorption, through the activation of ERK1/2, SGK1 and with-no-lysine kinase 4 (WNK4) signaling cascades. These results show that, in the context of low levels of Klotho, FGF-23 might be one of the explanations for CVD risks in patients with CKD [7].

Additionally, it is a point of interest that soluble Klotho regulates several processes, including anti-oxidation, anti-senescence, Wnt signal transduction and the anti-renin-angiotensin system (RAAS) [38]. It also inhibits fibrosis and apoptosis [51,52,53] and it affects mineral homeostasis through the regulation of FGF-23 and parathyroid hormone (PTH) secretion and phosphorus excretion by the kidneys, as will be discussed in Section 2.2.2 Klotho/FGF/PTH.

In summary, these findings support the relevance of Klotho/FGF-23 in CKD progression, as indicated in Figure 1.

#### 2.2.2. Klotho/FGF/PTH Axis

Populational studies have pointed out that, in CKD patients, the increase in FGF-23 and PTH, accompanied by the decrease in 1,25 dihydroxyvitamin D3, anticipate hyperphosphatemia [54], which is often observed in these patients and is usually related to a higher mortality risk among them [55,56,57]. Furthermore, hyperphosphatemia is frequently detected only when renal illness is irreversible and progressing to ESKD. It is important to mention that, alongside Vitamin D, PTH regulates not only calcium metabolism, but also phosphate metabolism [55,58], inducing phosphaturia [55,59]. Moreover, it stimulates the production of Vitamin D by the kidneys.

Data in the literature indicate that Klotho modulates PTH synthesis and release directly, and also through the regulation of the active form of Vitamin D and FGF-23 in plasma [12]. In addition, it has been proposed that there is a decrease in PTH production stimulated by FGF-23, when the expression of both FGF-23 and Klotho is normal in the parathyroid glands [60]. Interestingly, studies involving epidemiological data and animal models [61,62] have demonstrated that the decrease in Klotho expression at the beginning of CKD can lead to the overproduction of FGF-23, which results in secondary hyperparathyroidism, a common complication for patients with CKD [63], which contributes to other important comorbidities detected in this condition, such as CVD. It is worth mentioning that patients with CKD have high levels of FGF-23 in the blood [64] and lower expression of Klotho and fibroblast growth factor receptor (FGFR) one in the parathyroid glands [65], although Hofman-Bang, J., et al., have described the higher expression of Klotho in the latter organ [66].

Other models have been suggested to explain the relationship between Klotho and PTH. Imura, et al., for instance, proposed binding between Klotho and Na/K-ATPase in low Ca^2^+ levels, which would bring this transporter to the cell membrane and trigger the release of PTH, due to the change in the electrochemical gradient, although the exact signaling pathway involved in this model has not yet been completely elucidated [67].

A high level of FGF-23 is observed in patients and animal models with CKD [68], alongside a reduction in Klotho levels in the parathyroid glands [65,69,70]. The inhibition of PTH synthesis by FGF-23 is therefore lost [60]. In advanced stages of CKD, the low levels of Klotho and FGFR observed in the parathyroid glands lead to the inhibition of the suppressive activity promoted by FGF-23/Klotho signaling. Importantly, FGF-23—and Klotho as well—have been also considered important modulators for phosphate homeostasis controlled by the bone-kidney axis [55]. These data suggest that Klotho restoration may be an interesting approach to avoid the development of secondary hyperparathyroidism in CKD [12]. The administration of FGF-23 in animals with CKD, on the other hand, did not reduce the levels of PTH, which might result from low expression of both Klotho and FGFR1 in the parathyroid glands [60,71].

The levels of FGFR and Klotho, however, may differ among experimental models and stages of CKD, accompanied by differences in levels of calcium in the blood [12]. This fact highlights the need for a better illustration of how this axis works.

Razzaque, et al., demonstrated that FGF23-deficient mice develop hyperphosphatemia and a phenotype similar to aging and to Klotho-deficient mice [72], including vascular calcification related to hyperphosphatemia [1,73]. Hence, FGF-23 and Klotho are associated with phosphate homeostasis [55]. Interestingly, this phenotype can be reversed using interventions to reverse hyperphosphatemia [72,74,75,76]. These data suggest a link between phosphate levels and aging [55].

In short, studies have shown that both Klotho and FGF-23 [12] are able to regulate PTH synthesis. The hyperphosphatemia is believed to maintain the elevation of PTH levels in CKD and the dysregulation in the FGF-23/Klotho/PTH axis might lead to the progression of secondary hyperparathyroidism in CKD [12,38], as illustrated in Figure 2.

The instability in mineral metabolism, as described above, is a hallmark and also an initiator of the development of mineral bone disease in CKD [48], which contributes to higher mortality due to CVD and morbidity for these patients [77,78,79,80,81]. The FGF-23/Klotho axis, then, might be a target for new therapies for these patients [82].

#### 2.2.3. CKD and Cardiovascular Disease

Cardiovascular disease is an important and frequent morbidity in patients with renal illness and it is related to the mortality seen in these persons [83].

Preclinical and clinical studies have pointed out the relevance of Klotho, and also FGF-23, for the cardiovascular system and how it can be related to CVD in CKD individuals and mortality among the elderly and in hemodialysis patients [8,84]. The table below—Table 1—summarizes some of the results found regarding the topic.

In addition to the previously mentioned results, Faul, C., et al. evaluated the relation between FGF-23 and the pathogenesis of left ventricular hypertrophy (LVH) in humans, in patients from both the Chronic Renal Insufficiency Cohort (CRIC) study—a prospective cohort study conducted with CKD individuals [92]—and studies conducted by their group (Faul C and coworkers) [64]. A direct induction of hypertrophy in cardiomyocytes in vitro and in vivo (LVH) in mouse models has been demonstrated after FGF-23 administration. In mice lacking Klotho, which is in turn an accepted model for constitutively high FGF-23 levels, the development of LVH has been observed in a dose-dependent way. It is important to mention that the group indicated through molecular experiments that, although Klotho is the coreceptor for FGF-23, it is not expressed in murine heart preparations or neonatal rat ventricular myocytes (NRVMs). On the other hand, FGFR isoforms, from FGFR1 to FGFR4, were detected in vitro (NRVMs) and in vivo (murine heart); it is known that FGF-23 can bind these receptors [93,94], indicating that hypertrophy induced by FGF-23 in these sites is Klotho-independent. Moreover, the inhibition of FGFR with the intraperitoneal administration of PD173074 in a nephrectomy rat model ameliorated the severity of LVH, although improvements in CKD and hypertension were not observed. In brief, this study determined a causal association for elevated FGF-23 and the physiopathology of LVH, which could shed light on the high rate of LVH in CKD patients [64]. Regarding the onset of LVH, the same patients from this study were re-analyzed a few years later and it was found that high levels of FGF-23 were associated with an increased risk of LVH. Taken together, the results indicate that LVH can be preceded by high FGF-23 levels in CKD patients, both with or without hypertension [64].

A similar result was observed in a study conducted with nephrectomy CKD- induced rats, in which FGF-23 caused the hypertrophic growth of myocytes in vitro and induced LVH in mice, through the FGFR pathway [95].

In addition to the previously mentioned data, Klotho deficiency is also associated with vascular calcification (VC) in CKD, although the pathogenesis of this clinical outcome is not completely understood yet. It has been reported in a preclinical study involving mice and human aorta samples, for example, that the increase of Klotho in vitro led to the inhibition of VC [96]. Furthermore, the same research group from the previously mentioned study came up with a mechanism believed to modulate the expression of Klotho. They proposed that the mammalian target of rapamycin (mTOR) pathway is involved in the reduction of Klotho expression in a high-phosphate-concentration scenario and that the administration of rapamycin, an mTOR inhibitor, upregulates the levels of both membrane-anchored and soluble Klotho. This results in a decline in VC in vitro promoted by rapamycin, which was not observed in Klotho-knockout mice, indicating the pivotal role of this protein in the attenuation of VC evaluated in these models [96].

It is also interesting to mention that, as discussed in Section 2.2.2 Klotho/FGF/PTH axis, the lack of Klotho contributes to mineral disease in CKD for several reasons. Among them, Klotho deficiency results in a higher phosphate concentration in the organism, which leads to calcification of tissues as well [97]. Importantly, this imbalance in phosphate levels might also be related to the dysregulation in the PTH axis in CKD, believed to be associated with the lack of Klotho in the parathyroid glands [60,70,98].

Moreover, angiotensin II (Ang II) is also a special point of interest, because of its direct association with CVD, a possible complication in chronic kidney disease, and due to its pathological role, through several different mechanisms, as well as in CKD, as will be discussed. Reports have shown that Ang II and aldosterone suppress the expression of Klotho in the kidneys and in kidney cell lines [7,38]. Moreover, experiments with long-term infusion of angiotensin II in rats have shown a downregulation of renal Klotho mRNA, even with a low dose of Ang II. On the other hand, the overexpression of Klotho through in vivo gene transfer attenuates damages induced by Ang II in the kidneys [38]. Further analysis also indicated that losartan, an angiotensin type I receptor antagonist, blocked the reduction in Klotho levels induced by Ang II (in vivo and in vitro) and upregulated the expression of this protein in mice, improving structural alteration in the kidneys by nephropathy with cyclosporine. At the same time, one of the roles of soluble Klotho is anti-angiotensin activity in animal models. According to these results, evidence supports the association between Klotho and renoprotection from damages induced by Ang II [38].

The exact mechanisms by which Ang II reduces Klotho expression or contributes to kidney fibrosis are not completely understood yet, but experiments with free-radical scavenging have indicated the inhibition of Klotho expression by both Ang II and oxidative stress, which suggests that oxidative stress is one of these mechanisms. Furthermore, in a mouse model, inhibition of 1,25-(OH)2VD3 synthesis is correlated with increased renin expression, whereas its injection suppresses renin. Furthermore, a study conducted on vitamin D receptor (VDR)-null mice showed that they expressed higher levels of renin and Ang II. The same study has proved that, in mouse kidney and in HEK 293 cells, active vitamin D3 upregulates Klotho expression, reversing the decrease in Klotho’s mRNA by aldosterone. This demonstrates that supplementation with active vitamin D3 might have a positive effect for the upregulation of Klotho expression. Taken together, these results indicate that 1,25-(OH)2VD3 is a negative regulator of RAAS [38].

Additionally, Miao, J., et al., 2021 illustrated in mice a reverse in both mitochondrial loss of mass and the production of reactive oxidative stress with Klotho administration. Correspondingly, it is known that RAAS activation can lead to mitochondrial damage [99]. The inhibition of RAAS by Klotho, as shown in some prior studies [100], could protect mitochondrial function, contributing to the delay of age-fibrosis caused by RAAS. Although this effect has not yet been confirmed in the CKD context, this result suggests that it might be a possibility [99].

A current challenge in regard to RAAS inhibition is that the synthesis of Ang II is not affected by RAAS inhibitors (angiotensin-converting enzyme (ACE) inhibitors or Angiotensin II receptor type I (AT1) receptor blocker, for example) and there is also a stimulation of renin secretion as a consequence [99,101]. It is suggested, however, that multiple genes of RAAS are regulated by Wnt/β-catenin signaling. In this context, the inhibition of the Wnt/β-catenin signaling pathway has been proposed as a strategy for the inhibition of RAAS [41,99,101,102]. In accordance with this notion, Wnt-1, a component of the Wnt/β-signaling pathway, might be linked to kidney fibrosis, since studies with cultured tubular cells have shown that this molecule induced upregulation of fibronectin, a marker for fibrosis. This effect, though, was blocked by treatment with Klotho [99]. A point of interest is that Wnt/β-catenin activation may lead to Klotho expression. Experiments indicate that Klotho is an endogenous antagonist of Wnt and could prevent the activation of β-catenin through binding to other components of this signaling cascade, such as Wnt1, Wnt4 and Wnt7 [99]. This is one potential renoprotective mechanism for Klotho, considering the indirect inhibition of RAAS and the direct inhibition of Wnt/β-catenin. All in all, the information previously presented highlights the association between Klotho and cardiovascular disease in CKD—a common complication that often contributes to mortality in patients with CKD. This event has been observed through different mechanisms, as mentioned above, such as the increase in Klotho resulting from RAAS inhibition and the renoprotection conferred by this protein against renal damage induced by RAAS, whereas the activation of RAAS, on the other hand, is related to reduced levels of Klotho, for example. Moreover, decreased levels or a lack of Klotho are shown to be associated with vascular calcification, LVH and cardiomyopathy. FGF-23, in turn, is also related to endothelial dysfunction and LVH. Taken together, these data indicate the necessity of further research regarding Klotho and the Klotho/FGF-23 axis in CKD patients, since the results of present studies indicate the potential of these molecules as therapeutic targets to prevent mortality in these individuals, considering their involvement in the pathophysiology of a relevant complication in CKD.

#### 2.2.4. Klotho and Inflammation

As previously mentioned, CKD is a disease characterized by persistent inflammatory responses and cellular senescence [103]. The high levels of cytokines in this condition have been attributed by researchers to defective renal removal of these molecules, as well as their increased production and the inflammatory state itself in renal tissues of CKD patients [104]. In regard to this topic, Klotho has been associated with the inflammatory burden in CKD.

Indeed, studies have shown that NF-κB, one of the main proinflammatory molecules in CKD, downregulates the expression of Klotho in mouse models, and Klotho negatively modulates NF-κB, reducing the expression of this factor and the inflammatory state in CKD [7]. These results support the idea that some other proinflammatory cytokines are able to downregulate Klotho expression through an NF-κB dependent mechanism [7], highlighting the relevance of this molecule to the regulation of Klotho.

Furthermore, through NF-κB inhibition and consequent suppression of its target genes by Klotho [7,105], this protein abolishes endothelium adhesion molecules’ expression, such as intracellular adhesion molecule (ICAM-)1 and vascular cell adhesion molecule (VCAM-)1 induced by TNF-α in vitro [106]. Monocyte adhesion to cells, induced by TNF-α, was also abrogated by Klotho. Taken together, these data suggest that Klotho might participate in the modulation of vascular inflammation [106].

Another proinflammatory component involved with Klotho is TNF-related weak inducer of apoptosis (TWEAK), a soluble molecule that belongs to the family of TNF cytokines and is derived from its membrane-receptor enzymatic cleavage [107]. In vitro data have pointed out that TWEAK is able to activate NF-κB, leading to an increase in inflammatory cytokine expression, such as that of regulated upon activation, normal T-expressed and secreted (RANTES, also known as CC Motif Chemokine Ligand 5, CCL5) and monocyte chemoattractant protein-1 (MCP-1) [108]. Likewise, the activation of NF-κB and consequent cytokine expression by TWEAK has been also reported in vivo [109]. In rodent models, for example, Klotho expression is reduced with the injection of TWEAK [110]. Nevertheless, the inhibition of NF-κB suppressed this effect, which permits one to infer that the mechanism through which TWEAK decreases Klotho expression involves NF-κB [110]. In vitro experiments conducted by the same researchers from the previously mentioned study, in turn, have shown a downregulation of Klotho induced by TWEAK and TNF-α, whereas the expression of some inflammatory genes, such as IL-6, was higher. Again, this effect is believed to be related to NF-κB induction [110]. Interestingly, the inhibition of TWEAK could restore the levels of Klotho in AKI and preserve renal function [110].

Lastly, in addition to this information, studies with cultured rat aorta smooth muscle cells have indicated that Klotho transfection in cells resulted in a decline in NADPH-oxidase (Nox) 2 protein levels and a consequent reduction in oxidative stress in this in vitro model [111]. This approach also led to a rise in the levels of cyclic adenosine monophosphate (cAMP) and the activity of protein kinase A (PKA). The inhibition of this enzyme activity, on the other hand, abrogated the effect described above concerning Nox2 expression and Klotho. Thus, the study suggests that this modulation promoted by Klotho might rely on the cAMP-PKA pathway [111].

In summary, Klotho might be able to reduce the inflammatory state in CKD through the regulation of some inflammatory factors, such as NF-κB, and the expression of adhesion molecules as well as that of Nox2. Some of these factors, in turn, are also able to downregulate Klotho expression, as seen in the case of NF-κB and TWEAK. Thus, it is believed that the inflammatory state can also reduce Klotho levels, which could then result in negative impacts for the tissue, considering the potential anti-inflammatory activities that Klotho might have.

#### 2.2.5. Klotho and Fibrosis

Another relevant contributor to inflammation and the progression of CKD, and particularly to renal fibrosis, is TGF-β. Studies have shown both an upregulation of TGF-β and a downregulation of Klotho, followed by fibrosis, in rodent models with unilateral uretral obstruction (UUO), a model for CKD development [38]. It is also interesting to note that in mice with renal fibrosis induced by UUO, the inhibition of Klotho increased TGF-β1 expression; Klotho expression, however, not only attenuated pathological outcomes marked by inflammation, but also induced the decrease of fibrosis markers. In cultured renal epithelial cells, TGF-β1 was also able to decrease Klotho expression. These results indicate that Klotho deficiency can enhance TGF-β1 activity and that the former is not only a cause, but a result of CKD as well [38]. Another important finding was that in tubular cell lines of a rat model, Klotho suppresses the signal transduction induced by TGF-β1, thus inhibiting, mainly through inhibition of TGF-β1 signaling. When it comes to soluble Klotho, it has been observed, in mice with UUO-induced renal fibrosis, its binding to TGF-β type-II receptor and inhibition of TGF-β1 binding to cell surface receptors as a consequence. Thus, there was an inhibition of this signaling pathway. Furthermore, in renal epithelial cells, Klotho inhibited EMT. These results support the notion that Klotho can suppress renal fibrosis by inhibiting TGF-β1 activity. Experiments also indicated that Klotho supplementation can prevent and slow down CKD progression, attenuating the characteristic renal fibrosis in CKD [38].

## 3. Acute Kidney Injury

Acute kidney injury (AKI) is a disease with a sudden onset [112], marked by renal dysfunction that develops from a few hours to within seven days, according to the KDIGO Acute Kidney Injury Work Group. This disease is characterized by renal and extrarenal complications in organs such as the heart and the brain [113,114], due to the imbalance in electrolytes [115] and the accumulation of waste products [116,117,118]. In the kidneys, it can vary from minor renal deterioration to ESKD, especially in patients with CKD history, leading to dialysis [112,119,120]. Hence, this condition is associated with both high mortality and morbidity [116,121,122].

The worldwide prevalence of AKI is increasing and prior data in the literature have shown, for instance, that individuals who survive AKI can have a 28% rate of mortality in the first year after the onset of the disease [123], as well as a 50% increase in the risk of mortality during the period of approximately 10 years of follow-up [124,125]. These patients might face other long-term outcomes, such as a higher risk of CKD [126], although the exact mechanisms for this process are not yet well elucidated [127]. Furthermore, patients in intensive care units (ICU) have a 50–70% rate of AKI [116,128,129]; therefore, this disease is considered one of the most worrying issues for hospitalized individuals in ICUs [117,130,131,132]. As expected, acute kidney injury also has an impact on costs in the health system [133,134].

This syndrome has diverse etiologies, such as other prior conditions, such as sepsis, acute or chronic illnesses [135], ischemia-reperfusion injury (IRI) and even the use of nephrotoxic drugs [115]. Likewise, the aging of the population represents another important trend to a higher incidence of AKI [136].

The standard diagnosis for AKI includes an increase in either serum creatinine of 0.3 mg/dL or more within forty-eight hours, in serum creatinine by 1.5 times within seven days or a urine volume inferior to 0.5 mL/kg/h for six hours [117,137]. A reduction in the glomerular filtration rate (eGFR) [138] is also a common consequence of this condition.

There are a variety of pathological processes associated with AKI, although the exact mechanisms involved in its physiopathology are not yet completely understood [139,140]. Proximal tubule cells are the main cells affected after a nephrotoxic insult [141] and inflammation is an important response for the development of AKI [142,143,144]. Proinflammatory molecules are released by renal and endothelial cells, resulting in an infiltration of inflammatory cells [145]. As a consequence, there is damage to renal tubules, characterized by cell death via necrosis and apoptosis, cytoskeleton disruption [115] and oxidative stress [146], for instance. Moreover, after a severe injury in the kidneys, tubular fibrosis and a senescent-like phenotype in tubular cells [147] might occur. The upregulated production of profibrotic factors, such as TGF-B, leads to the activation and proliferation of fibroblasts [148,149], stimulating the production of extracellular matrix and tubulointerstitial inflammation [149,150,151]. There are also other proinflammatory molecules that contribute to the progression of AKI, such as NF-κB [152,153,154].

As previously mentioned, severe AKI is considered an independent and important risk factor for the course of CKD [124,125,155,156]; as such, patients who experience AKI are more likely to present either CKD or ESKD [157,158]. Likewise, patients with CKD might also have transient states of renal dysfunction corresponding to AKI [157].

In general, the renal impairment caused by this disease is reversible, although long-term outcomes might exist, as previously discussed [159]. In addition, AKI is associated with prolonged hospitalization. There is, however, a lack of efficient therapies for AKI currently and few biomarkers that are representative of the early stage of the disease have been used in clinical practice [160,161,162,163]. Thus, both prevention and treatment of this condition are of pivotal importance [115,145].

In this context, Klotho has been suggested to be possibly related to AKI, as will be addressed next.

### 3.1. Klotho in Acute Kidney Injury

Studies have demonstrated that Klotho production is reduced in different models of AKI, such as in cisplatin-induced AKI [146] and AKI induced by IRI [164], which contributes to kidney damage during this disease [46,145,165]. Hu et al. reported, for instance, in a preclinical and clinical study, that in rodents with IRI-induced AKI, the levels of Klotho were reduced in the kidneys, urine and blood. Moreover, they also showed that a decrease in this protein level occurred before the reduction of other early biomarkers for kidney injury [46]. In AKI patients, researchers also detected a reduction of urinary Klotho levels, compared to healthy individuals [46].

In order to evaluate the role of Klotho in AKI, the same research group induced different levels of this protein in mice. A higher resistance to injury was found in rodents with a higher expression of Klotho; hence, these animals displayed fewer kidney alterations, which indicates that the overexpression of Klotho might mitigate AKI, whereas its deficiency accentuates the disease [46]. Concerning the restoration of Klotho levels, in rats, the administration of recombinant Klotho led to less renal damage in comparison to the group with no such treatment. Furthermore, it has been shown that the earlier the injection of Klotho after ischemia, the more effective this approach is to improve kidney conditions in AKI [46]. Other studies conducted with animal models also point out the relevance of this strategy in improving renal fibrosis and pathogenesis of AKI [146].

Taken together, these results provide evidence that there is indeed a deficiency of Klotho in AKI and that this contributes to renal damage. Moreover, they also highlight that this protein is both an early biomarker [38,46] and a contributor to AKI pathogenesis [146], and it is thus possible to study it as a potential therapeutic tool, considering that renal injuries are attenuated upon administration.

The exact mechanisms through which Klotho influences AKI and is downregulated are not well elucidated yet, though; some of these will be described below. It is important to note that there are several different models for the study of AKI; cisplatin-induced renal injury is a widely accepted one [145] and Klotho is reduced in this model [164]. This protein level, however, is also reduced in other models of AKI, such as in IRI [46,146], AKI induced by LPS and folic acid [110].

#### 3.1.1. Klotho, Inflammation and AKI

During the development and progression of AKI, there is a dysregulation in cellular processes; some of them are related to Klotho. Data have indicated that the downregulation of Klotho in AKI is associated with cellular senescence and, importantly, that this process might be induced as a response to oxidative stress [166], which is a contributor to inflammation. Moreover, reports have shown that Klotho protects the kidneys, having an anti-oxidative [167,168] role, since it can stimulate the expression of antioxidant enzymes [169]. These data are supported by the fact that a deficiency of Klotho was also shown to be associated with increased levels of oxidative stress [170] in IRI models of AKI. Furthermore, experiments involving H_2_O_2_ as an insult similar to IRI in rodents have highlighted that co-incubation of cells with Klotho reduced the release of lactate dehydrogenase (LDH) [171]. It has been reported by Sun, M., et al., 2019, for instance, in a study involving septic mice with AKI, that Klotho has a renal protective role and the mechanism for this process is related to the maintenance of mitochondrial integrity and protection against oxidative stress [172].

Furthermore, Bi, F., et al., 2018, observed that Klotho is able to suppress lipopolysaccharide (LPS) AKI through the degradation—via deglycosilation—of toll-like receptor (TLR) 4 [173].

Regarding NF-κB in studies conducted in mice, TWEAK and TNF-α were responsible for the downregulation of Klotho, and it has been reported that the mechanism involves NF-κB [110]. Furthermore, data also suggest that NF-κB is able to suppress Klotho expression through association with histone deacetylase (HDAC) 1 and nuclear receptor corepressor (NCoR), which interacts with Klotho promoters and may repress its transcription in inflammatory conditions. Moreover, in the same study, the researchers present further evidence of the importance of the anti-inflammatory effects of Klotho in AKI [174]. Klotho silencing leads to an aggravated inflammatory response in a rhabdomyolysis model, causing higher expression of TNF-α and IL-1β, when compared to control mice injected with siRNA-control. Interestingly, Klotho is of pivotal importance for the renoprotective effects of nicotinamide, the active form of vitamin B3, which prevents NF-κB and corepressors recruitment to Klotho promoter, therefore attenuating inflammation and rhabdomyolysis-induced AKI and preserving Klotho expression [174].

Hence, Klotho is seen as a potential anti-inflammatory molecule in AKI [168,170,173], due to its association with NF-κB and its protective role against oxidative stress, for example. However, it is worth mentioning that there is a scarcity of data in the literature relating to the exact mechanisms through which Klotho is associated with inflammation in AKI, which highlights the importance of further studies regarding this topic.

#### 3.1.2. Klotho and Non-Inflammatory Mechanisms in AKI

In addition to the inflammatory aspects mentioned above, there are also non-inflammatory events associated with Klotho and AKI.

Concerning its anti-fibrotic function, Klotho is an endogenous Wnt antagonist, blocking, as a result, the activation of β-catenin. Through the inhibition of this cascade, as previously mentioned, by binding to Wnt ligands (such as Wnt1 and Wnt4), increased levels of Klotho can reduce fibrosis in kidneys and ameliorate renal function [100,175]. In animal models of AKI, such as unilateral ureteral obstruction, restoration of Klotho can avoid renal fibrosis [176,177]. Prior studies have also shown that, in mice, Klotho inhibits Smad signaling induced by TGF-β; as a result, it interrupts fibrotic signaling. There are other mechanisms through which Klotho exerts an anti-fibrotic function in the kidneys, such as the inhibition of HDAC. This inhibition is causally affected by Klotho and contributes to bone morphogenetic protein 7 (BMP-7) restoration, a protein that has a renal protection role by promoting the repair and proliferation of cells from renal tubules cells after injuries [178]. The downregulation of BMP-7 worsens renal complications [179,180]. Furthermore, data from an UUO mice model of AKI demonstrated that the administration of soluble Klotho was able to suppress fibrosis, through binding to the type II receptor of TGF-β, which inhibits TGF-β signaling [177].

Moreover, studies have indicated that there is cycle arrest in AKI [147]. Researchers have demonstrated a positive correlation between G2/M cell cycle arrest and the synthesis of cytokines related to the fibrotic process in tubular cells, through a c-jun NH(2)-terminal kinase (JNK) signaling pathway [147]. Cell cycle arrest may also lead cells to senescence. Klotho, in turn, was shown to be protective against cell senescence after AKI induction [181]. Studies have shown, for instance, the attenuation of apoptosis and senescence in cultured endothelial cells after Klotho recombinant protein administration, through mitogen activated-kinase kinase (MAPK) and ERK signaling pathways [181]. Furthermore, in vivo studies with mice show that the overexpression of Klotho can abrogate senescence phenotypes in injured renal tissue. In animals with a high expression of Klotho, there is also a reduction in mitochondrial DNA damage, which has been attributed by researchers to this protein. Moreover, researchers have observed a decrease in oxidative stress when Klotho is overexpressed [47].

There is also evidence of increased Wnt signaling pathway activation in animal models that are deficient in Klotho. This event is associated, in vitro and in vivo, with cellular senescence [182]. Thus, Klotho has been associated with the inhibition of the Wnt/β-pathway in AKI, and it has therefore been considered as an anti-fibrotic molecule.

In mouse models for AKI induced through IRI, in turn, studies have demonstrated that the delivery of Klotho leads to improvements in apoptosis, histological damage and creatinine values. It has been reported that there is an increase in HSP (heat shock protein) 7o expression according to Klotho levels in this animal model, which contributes to the amelioration of apoptosis [183]. Hence, Klotho has also been shown to be involved with the reduction of both senescence and apoptosis, along with the improvement of renal parameters, in different models of study involving AKI.

Moreover, autophagy is activated in several models of AKI, such as in unilateral ureteral obstruction [184,185], IRI [186,187] and cisplatin-induced AKI [188,189,190,191,192], and different studies have shown that lower activity of this biological process can lead to vulnerability in ischemia and nephrotoxicity [184,188,189,190,191,192,193,194]. Likewise, balanced autophagy activity in the kidneys protects them from several renal insults [195,196,197,198]. Interestingly, Klotho has been shown to be associated with autophagy levels [199,200,201,202], although the exact molecular mechanism of this association is not yet completely understood.

Experiments with transgenic mice overexpressing Klotho have shown a positive correlation between higher autophagic flux and higher Klotho levels; at the same time, renal cells can become more vulnerable to oxidative stress when autophagy is suppressed. Furthermore, in cell culture experiments, autophagy inhibitors resulted in a decrease of Klotho’s induction of autophagy and cytoprotective effects against H_2_O_2_. It can inferred by these results that autophagy is one of the processes through which Klotho induces protection in renal cells, since its activity is upregulated by this enzyme [203].

Further analysis has also shown that autophagy contributes to the maintenance of collagen balance in renal cells, according to experiments involving autophagy inducers and inhibitors to evaluate the expression of collagen type I (Col I) in OK cells. Likewise, transfection of OK cells with Klotho resulted in a decrease in Col I accumulation, both extracellular and intracellular. As this effect was abolished in part with the use of autophagy inhibitors, it has been suggested that this regulation of Col I promoted by Klotho might depend on autophagy flux [203].

Moreover, there is evidence that due to the collagen degradation stimulated by autophagy [184,204,205] and the induction of autophagy by Klotho, the amelioration of fibrosis in the kidneys promoted by Klotho might be associated with autophagy. Experiments in cell cultures with autophagy inhibitors have shown a decrease in Col I degradation by Klotho [203].

Taken together, these results suggest that autophagy is one of the mechanisms through which Klotho protects the kidneys.

## 4. Therapeutic Potential of Klotho in Acute and Chronic Kidney Diseases

The reestablishment of Klotho provides benefits concerning the progression of renal disease and other events associated with it, as it has been discussed in this review and as briefly shown in Figure 3.

Given the importance of Klotho and its deficiency in both CKD and AKI, some studies have been conducted into strategies for elevating its levels as a therapeutic approach. Buchanan, S., et al. have reviewed studies concerning some possible interventions to increase the levels of Klotho [2]; some of these are summarized in Figure 4.

Concerning the approaches described in Figure 4, it is important to explore the advantages and also the negative aspects concerning each one of these approaches, as addressed in the next paragraphs of this review.

(a).DNA methyltransferase inhibitor: In regard to the DNA methyltransferase inhibitor azacytidine, it has been observed that Klotho’s promoter is located in a region rich in cytosine and guanine—a CpG island [206,207] that lacks sequences for classic regulatory elements in this region [206]. Azacytidine, in turn, is able to promote an augmentation in the promoter activity of the Klotho gene, leading to a rise in the levels of this protein in cells. The use of this compound in vivo, though, is difficult and regards future research due to the variety of possible activities presented by this compound [2].(b).Agonists for PPAR-γ: Moving on to troglitazone and ciglitazone, these drugs are classified as thiazolidinediones and they act as agonists of peroxisome proliferator-activated receptor gamma (PPAR-γ) [208]. One study indicated that, in cells from medullary collecting ducts, proximal tubules and distal tubules, there is an induction of the expression of Klotho genes by these compounds, both in a time- and dose-dependent manner. The same study proposed that the activation of this receptor is, then, a potential mechanism for the effect observed, since a selective PPAR-γ antagonist abolished the process [208]. In regard to the clinical use of these molecules, the challenges involve edema, weight gain and osteoporosis, among others [209].(c).Histone deacetylase inhibitors: The other approach discussed in Figure 4 is the use of histone deacetylase inhibitors, such as trichostatin A and valproic acid. Data indicate that in cells treated with these compounds, the reduced expression of Klotho induced by TWEAK or TNF-α is prevented [110]. Likewise, an increase in Klotho’s gene expression in some cell lineages has been reported [206]. In spite of the fact that these inhibitors are being evaluated in clinical trials involving the treatment of cancer, for example, their use in vivo is still difficult, because of adverse reactions, especially cardio-toxicity [210].(d).RAAS inhibitors: Regarding the use of RAAS inhibitors, in turn, it has been demonstrated in a rodent model of chronic nephropathy that the inhibition of angiotensin II type 1 receptor with the use of losartan results in an augmentation in Klotho expression, along with a reduction in kidney histological damage [211]. A meta-analysis of randomized controlled trials and systematic reviews, however, has shown that in order to improve some functional parameters in CKD patients, such as blood pressure control and the reduction of proteinuria, dual blockade of RAAS is better than monotherapy [212]. Thus, although the previously discussed alternative represents a potential strategy for the treatment of CKD, the approach itself does not diminish the development of ESKD, for example, so it does not cause a long-term amelioration in the treatment of CKD, as reviewed by that group [212].(e).Paricalcitol: As illustrated in Figure 4, other compounds are interesting for the elevation of Klotho levels, such as vitamin derivatives, such as paricalcitol. In a study with uremic rats, conducted by Ritter, it was demonstrated that paricalcitol blocks the reduction of Klotho mRNA and protein levels in renal tubules [213]. In another study, on the other hand, an increase in Klotho levels in the kidneys was not observed [214]. Currently, there is a lack of information in the literature in regard to the influence of paricalcitol and other agonists for vitamin D receptors on CKD progression and cardiovascular risk [215]. Furthermore, a meta-analysis highlighted the necessity of future randomized trials to assess the effects of paricalcitol on ESKD progression and mortality in individuals, although some data point out that this approach is effective in the reduction of proteinuria [216]. Moreover, the authors reported that there was a trend towards hypocalcemia in patients [216]. Thus, the clinical feasibility of this approach still needs further studies.(f).Intermedin: In regard to intermedin, a study with rats with CKD has shown that the decrease in Klotho protein levels was overturned by intermedin in the kidneys, plasma and calcified aorta [217]. This compound also diminished vascular calcification in this animal model [217]. Intermedin, then, could be a promising strategy to increase Klotho levels. However, the lack of information in literature about this compound is a downside of it. Only two experimental papers were found when we searched for information about intermedin and Klotho in Pubmed [217,218]. Thus, further studies are needed in order to shed light on the efficacy and safety of intermedin and on how exactly it is related to Klotho levels.(g).Statins: Statins have also been proposed as a promising approach to promote Klotho overexpression. Studies with rodents demonstrated that these drugs led both to an increase in renal Klotho levels and to the attenuation of the reduction of Klotho in nephropathy [219]. Furthermore, there was an improvement in the resistance to oxidative stress in CKD [219]. Although the use of statins for patients with correct medical recommendations is considered acceptable because of their benefits [220], a review has pointed out that genetic testing of transporter genes which affect the internalization or efflux of statins would be interesting, considering the existent polymorphisms that can affect both the safety and efficacy of statins, since this genetic background might interfere in the incidence of myopathy and statins results in patients [221].(h).Recombinant protein: The administration of recombinant Klotho, in turn, is a potential strategy to increase its levels [48]. Studies with rodent models for kidney diseases have shown the attenuation of renal fibrosis [177], the avoidance of progression from AKI to CKD, cardiac remodeling and an improvement in renal and cardiac parameters [177]. Researchers have also observed the reduction of kidney damage and recovery from AKI [46]. However, the Klotho protein’s instability in urine and blood [222] might demand alternatives to an effective recombinant protein administration [223]. Timing for treatment should also be analyzed so that there would not be an impaired recovery of the kidneys [223].(i).Adenoviral delivery: Lastly, as described in Figure 4, adenoviral delivery of Klotho leads to an increase in this protein level. Studies with rodents with reduced Klotho expression have indicated the mitigation of renal damage promoted by adenoviral Klotho delivery, seen, for example, with a decrease in tubular atrophy [224]. Regarding this approach, there are also other studies involving rodent models for diabetic kidney disease (DKD) that show the amelioration of the disease after the delivery of Klotho, which might be associated with the inhibition of RAAS activation and the Wnt/β-catenin pathway [225] and the improvement of creatinine clearance, proteinuria and tubulointerstitial damage in animals with Ang-II infusion [226]. Moreover, in rodent models for AKI studies have demonstrated the mitigation of histological damage and apoptosis and the improvement of serum creatinine levels post-injury after Klotho delivery [183]. Although promising results have been achieved so far, it is important to mention that the adeno-viral delivery of genes requires caution, due to possible mutagenesis or immunogenicity [227].(j).Mesenchymal stem cells and extracellular vesicles: Taking into account all the approaches discussed previously, we decided to explore the potential of MSCs due to their applicability in regenerative medicine. Currently, these are the most studied type of stem cells [228] and they are also the most commonly used ones [229]. They are seen both as a therapeutic approach themselves, through their direct administration to patients—either genetically modified or not—and as a way to grow organoids in culture, for instance, which can also be administered to patients later [230]. These cells present potential for the treatment of different diseases, due to their immune modulation and differentiation properties, along with their tropism for injured tissues and their rich secretome, as it will be addressed in this review [229]. Hence, considering these characteristics of MSCs, alongside Klotho’s anti-inflammatory potential, we speculate that these cells can act like a “Trojan Horse’’ in vivo, delivering Klotho to the renal tissue, and that they could therefore modify and improve the microenvironment conditions for Klotho. This protein, in turn, would be able to ameliorate the surroundings conditions for MSCs as well, and consequently increase their efficacy in vivo. Therefore, we propose that MSCs and Klotho would act in a synergic way, contributing to the improvement of kidney conditions in CKD and AKI, as illustrated in Figure 5.

Taken together, these data reveal that Klotho can be used both as a biomarker and, potentially, as a therapeutic tool for renal diseases [46].

### 4.1. Mesenchymal Stem Cells

#### 4.1.1. Properties and Characterization of Mesenchymal Stem Cells

Mesenchymal or stromal stem cells (MSCs) are cells found ubiquitously in the perivascular region. As we have previously discussed, these cells were first described nearly 50 years ago and extensive research has documented their potential to differentiate into the three germ layers in vitro [231]. Despite this multipotent capacity, there is compelling evidence that MSCs contribute to tissue regeneration through paracrine effects, in particular via secretome production, and their direct differentiation in vivo into damaged cells in a specific organ is not relevant from a clinical perspective [231].

The MSC-derived secretome comprises extracellular vesicles (EVs), such as microvesicles and exosomes that differ in their size (50–1000 nm vs. 30–150 nm), origin (outward budding and shedding from plasma membrane vs. endolysosomal pathway and exocytosis) and cargo (surface proteins, cytosolic proteins, DNA, mRNA and miRNA vs. exosomal proteins, cytosolic proteins and miRNA) [232]. EVs play a crucial role in cell–cell communication, in particular the immunologic response, angiogenesis, cell proliferation and cell differentiation. Furthermore, MSCs were able to prevent tissue damage through the transfer of mitochondria [233].

The MSC-derived secretome represents is a cell-free alternative for treating kidney diseases, which can bypass some issues related to autologous and allogeneic MSCs [234]. This secretome can be obtained from different sources of MSCs, including bone marrow, adipose tissue, the umbilical cord and Wharton’s jelly. Some advantages include their time-saving nature, scalability, the absence of antigenic factors, their ease of obtainment and the adaptation of MSC to producing preestablished secretome components, designed to target specific diseases, and even allowing the separation of vesicles from soluble proteins and the management of the cell product according to each disease setting.

Moreover, EVs isolated from MSCs are promising strategies in kidney diseases, as they modulate several biological processes, such as oxidative stress, apoptosis, inflammation, fibrosis, angiogenesis, cell cycle, regeneration, autophagy and cell proliferation [235,236].

Strategies for targeting EVs to particular target cells can also be achieved through genetic modification. Therefore, these EVs can also be engineered and loaded with Klotho recombinant protein or other cargo for kidney repair, as previously described for urine-derived EVs [237]. Importantly, microvesicles isolated from BM-MSCs can be integrated into tubular epithelial cells and lead to an improvement in morphological and functional parameters in glycerol-induced AKI [238].

Other genetic modifications of EVs comprise the modification of these EVs to express targeting moieties fused with exosome-native membrane proteins, such as lysosomal-associated membrane protein 2 (Lamp2b), tetraspanins, glycosylphosphatidylinositol (GPI) and lactadherin C1C2, as well as engineering exosome-liposome hybrids [239,240].

Taken together, these findings shed light on the use of engineered EVs from MSCs for Klotho delivery in acute and chronic kidney diseases.

However, EV-based therapy imposes some challenges, which include the recovery of EVs, purity, specificity, sample volume, efficiency, complexity, the functionality of EVs, time, cost, the need for advanced equipment and scalability [241]. For large-scale production, tangential flow filtration (TFF) combined with a size-exclusion chromatography (SEC) column efficiently yields high-purity EVs [240] and should be pursued for clinical applications. MSCs are immune-privileged cells, as they do not express class II MHC (major histocompatibility complex) or stimulatory molecules such as CD86, CD40 and CD40L, which has prompted the development of clinical studies of not only autologous but also allogeneic cells in several acute and chronic diseases. Therefore, for therapeutic purposes, the characterization of MSCs should include the following in vitro properties, according to the International Society for Cellular Therapy (ICST, 2006): (a) adherence to plastic; (b) immune-phenotype profile with positivity for CD29, CD44, CD73 and CD105, as well as negativity for hematopoietic markers (CD34 and CD45), monocyte markers (CD14 or CD11b) and lymphocyte markers (CD79 or CD19, as well as HLA-DR); (c) multipotent capacity to differentiate into mesoderm-derived cells, in particular adipocytes, chondrocytes and osteocytes [242].

Furthermore, in order to standardize the characterization of MSCs and reduce their heterogeneity, the updated version of the ISCT guidelines (2016) recommended the following analyses of the properties of MSCs before they are administered for therapeutic purposes: (a) self-renewal capacity and clonogenicity assessed via the colony-forming unit fibroblast assay (CFU-f), which means that each colony arises from a single precursor cell, and the doubling-time; (b) analytical methods for measuring potency, including extensive product characterization data evaluating immunochemical (quantitative flow cytometry or enzyme-linked immunosorbent assay), biochemical (protein binding or enzymatic reactions) and/or molecular (reverse transcription polymerase chain reaction, quantitative polymerase chain reaction or microarray) attributes of the product; (c) functionality assays focusing on the specific therapeutic properties that are sought in vivo, in particular the angiogenic potential and the immunomodulation assays performed with MSCs and the lymphomyeloid cell population; (d) assessment of immune plasticity at both phenotypic and functional levels; and (e) karyotype [243]. When the MSC-derived secretome is analyzed, it is therefore possible to identify a molecular signature upon exposure to various pro-inflammatory cytokines such as interferon (IFN)-γ, tumor necrosis factor (TNF)-α, IL-1α and IL-1β [244]. These pro-inflammatory cytokines mimic the microenvironment of dysregulated immune responses or systemic inflammation found in patients during MSC infusion, as the measurable immunological characteristics of MSCs depend on their activation status at the time of interaction with effector cells. Thus, this functional assay recapitulates the immune regulatory functions of MSCs, since their phenotype under resting conditions and their polarization toward inhibitory functionality after pro-inflammatory stimuli can be evaluated from different donors, as well as from MSCs from different sources.

Additionally, the ISCT recommends including appropriate reference materials, standards and/or controls and establishing and documenting the accuracy, sensitivity, specificity and reproducibility of the test methods used through validation on a regular basis in facilities with Good Manufacturing Practices (GMP) [243].

However, there are some challenges when clinical outcomes are compared across studies, such as the MSC source, which most frequently includes bone marrow, adipose tissue and the umbilical cord; the route of administration, which most often includes intravenous, followed by intra-articular, intracardiac or intrathecal depending on joint, cardiovascular or neurological diseases, respectively; the number and/or frequency of cell administration; and the homing capacity and severity of the patient’s condition [245,246].

#### 4.1.2. Efficacy and Safety of Mesenchymal Stem Cells

In a kidney disease setting, MSC efficacy is challenged by several aspects. Therefore, the inflammatory milieu in the course of kidney disease can be modified and the MSC phenotype is modulated accordingly [247,248]. When the environment is non-inflammatory (low levels of IFN (interferon)-γ and TNF-α), MSCs acquire a proinflammatory phenotype (MSC1), and when Toll-like receptor (TLR)-4, LPS (lipopolysaccharide) and high levels of chemokine C-X-C motif ligand (CXCL)9, CXCL10, MIP (macrophage inflammatory protein)-1α, MIP-1β and CCL5/RANTES (regulated on activation, normal T cell-expressed and secreted), but low levels of IDO (indoleamine 2,3-dioxygenase), NO (nitric oxide) and PGE2 (prostaglandin E2) are detected, the activation of cytotoxic T lymphocytes is triggered. In contrast, in an inflammatory environment associated with high levels of IFN-γ and TNF-α, MSCs acquire an immunosuppressive phenotype (MSC2) and through TLR-3 lead to an increase in the production of TGF-β, IDO, NO and PGE2. These events stimulate the amount of CD4^+^CD25^+^FoxP3^+^ T regulatory cells and may explain the immunomodulatory properties of MSCs and their involvement in tissue regeneration.

Additionally, inflammation may also affect the interaction of MSCs and monocytes. When MSCs acquire an immunosuppressive phenotype in the presence of IL-6, they produce high levels of IDO and PGE2, which contributes to polarization from monocytes (M0) to a macrophage anti-inflammatory phenotype (M2 macrophages, which are characterized by CD206 and CD163 expression and the production of high levels of IL-6 and IL-10). Conversely, the proinflammatory MSC-induced phenotype may lead to polarization from M0 to proinflammatory macrophages (M1 macrophages; which are characterized by CD86 expression and the production of high levels of IFN-γ and TNF-α) [247,248].

Therefore, the clinical condition of the recipient also plays a key role in our understanding of the mechanisms of MSC-mediated tissue regeneration. Thus, in individuals with diabetic ketoacidosis, stem cell therapy lacks efficiency in controlling hyperglycemia [249]. The burden of chronic diseases, such as DM, which is the leading cause of CKD globally, may ultimately affect the potency of MSCs. Importantly, type 2 DM-derived AT-MSCs exhibit a lower proliferative capacity, higher levels of senescence and apoptosis, and a decreased potential for differentiation when compared to healthy donor-derived MSCs [250,251]. Likewise, MSCs isolated from type 2 DM produce lower levels of vascular endothelial growth factor A (VEGFA) and C-X-C chemokine receptor 4 (CXCR4), which compromises their angiogenic and migratory potential [251]. These findings have biological implications, as autologous MSCs from type 2 diabetic individuals may not function properly. On the other hand, type 1 DM-derived MSCs, although exhibiting a distinct gene profile from healthy donors, possess an equivalent colony-forming unit-fibroblast capacity, cytokine secretion, immunomodulatory activity and wound healing potential [252]. The difference between type 1- and type 2-derived MSCs may be at least in part explained by the older age and the number of comorbidities in the latter.

In line with these findings, MSCs obtained from individuals with cardiovascular disease (CVD), the leading cause of death in CKD, show increased senescence and reduced proliferation and regenerative potential, despite studies presenting conflicting results [253]. However, DM and CVD seem to have an additive effect in decreasing MSC potency. Therefore, the characterization of MSC function accordingly to ISCT is of paramount importance to guide decisions regarding the use of allogeneic or autologous cells.

In terms of alloimmune reactions in individuals receiving allogeneic MSCs, it was documented that these responses were very low [254,255].

Moreover, the route of MSC delivery plays a crucial role in MSC efficiency. As previously reviewed, the intravenous route is chosen more frequently for treating kidney diseases due to the relative ease of the procedure, even though MSCs are initially trapped inside the pulmonary vasculature. Intra-arterial routes, including intra-aorta, intra-renal artery and intracarotid routes, are associated with a more robust repair of kidney injury. Although the intraparenchymal route, e.g., under the renal capsule, leads to kidney improvement, this route is less practical for clinical applications [256].

The cell source, for instance, is a great subject of debate nowadays. Bone marrow is the most used MSC source, followed by the adipose tissue and the umbilical cord [257]. The MSCs isolated from these sites may present different in vitro and in vivo behaviors that may affect their application. Bone marrow MSCs (BM-MSCs) exhibit a lower proliferation rate when compared to the other sources and also express a secretome profile that enhances its osteogenic and condrogenic potential, as BM-MSCs secrete higher amounts of stromal cell-derived factor (SDF)-1 and HGF [257,258,259,260]. However, AT-MSCs are easier to isolate due to high availability and accessibility. Moreover, AT-MSCs have higher adipogenic potential and a secretome composed of high levels of insulin-like growth factor (IGF-1), VEGF-D and IFN-γ [259,261].

It is not only MSC characteristics that are involved in kidney repair. The evaluation of its therapeutic potential in the most appropriate pre-clinical model that mimics renal disease in humans is also of paramount importance for advancing our understanding of MSC-based therapy. Therefore, a large body of evidence has demonstrated the impact of this therapy in acute kidney injury, including IRI, kidney transplantation and drug-induced kidney injury [262], as well as chronic kidney disease, in particular DKD [256]. Whether the same kind of results is seen in humans with multiple comorbidities remains to be studied in detail.

One of the most important aspects of advancing MSC-based therapy is defining the MSC source, whether they are derived from allogeneic or autologous donors. As we have previously reviewed, aging and chronic diseases, in particular diabetes mellitus and DKD, may decrease the potency of MSCs, including a decrease in their homing capacity and potential for angiogenic differentiation [256]. These findings point to the development of MSC biobanks, since alloimune reactions in patients treated with allogeneic MSCs are very low (3.7%) [254].

Therefore, MSCs contribute to mitigating acute kidney injury after ischemic and toxic injury [262], and to curtailing chronic kidney disease, including DKD [263], 5/6 nephrectomy [264] and renovascular hypertension [265]. In the kidney disease setting, MSCs have contributed to immunomodulation by reducing the levels of pro-inflammatory cytokines such as IL-1β, IL-6 and TNF-α, as well as inhibiting in macrophage infiltration by decreasing the expression of monocyte chemoattractant protein-1 (MCP-1) and also increasing the levels of anti-inflammatory interleukin IL-10 and indoleamine 2,3-dioxgenase (IDO), which contributes to stimulating regulatory T cells; secreting trophic factors (hepatocyte growth factor (HGF), epidermal growth factor (EGF), vascular endothelial growth factor (VEGF) and bone morphogenic protein-7 (BMP-7)); ameliorating oxidative stress (decreasing levels of malondialdehyde (MDA), reactive oxygen species and protein carbonyl and increasing superoxide dismutase); decreasing fibrosis (decreased expression of TGF-β1, PAI-1 or plasminogen activator inhibitor-1, collagens type I and IV and α-SMA or α-smooth muscle actin); and modulating kidney injury (increased expression of nephrin, podocin, WT-1, synaptopodin, megalin, zonula occludens-1 and E-cadherin, as well as tubular Ki67 proliferation index and anti-apoptotic factors, including Bcl-2, and the reduction of pro-apoptotic factors such as BAX) [256,262]. By attenuating kidney damage, MSCs may lead to better kidney function, including a reduction in serum creatinine, urea and albuminuria, ultimately leading to animal survival. Notably, adipose-tissue-derived MSCs (AT-MSCs) injected into animals with streptozotocin-induced renal injury, e.g., DKD, prevented the progression of kidney disease by decreasing Bax and Wnt/β-catenin levels, and increasing Bcl-2 and Klotho levels [266].

These findings have important clinical and pathophysiologic implications regarding the impact of MSC-based therapy on the burden of kidney diseases. As kidney diseases encompass the dysfunction of a wide range of cells, including epithelial tubular cells from distinct compartments, interstitial cells, endothelial cells, mesangial cells and podocytes, strategies to enhance MSC potency are further warranted. Therefore, preconditioning of these cells with hypoxia, pharmacological agents, trophic factors/cytokines, physical factors/material and small molecules, as well as genetic modification and the optimization of MSC culture conditions, are currently being pursued [267]. These strategies are effective in activating several signaling pathways that are important in protecting cells and organs from injury, in particular in reducing apoptosis, oxidative stress and fibrosis, as well as in increasing cell proliferation and migration and angiogenesis. Therefore, preconditioning strategies may ultimately reduce MSC heterogeneity and represent a key approach to setting the basis for advancing cell therapy in preclinical and clinical studies.

Importantly, preconditioning of MSCs in a hypoxic environment with 2–5%, which mimics the MSC environment, allows these cells to remain multipotent and have a greater proliferative and migratory capacity, and exhibit low rates of senescence [268,269]. Notably, hypoxia-preconditioned MSCs do not differentiate into tumors either in vivo or in vitro [268]. Additionally, preconditioning MSCs via stimulation with TNF-α, IFN-γ, PGE2 and nitric oxide leads to more homogenous behavior by MSCs in T lymphocyte proliferation assays and late-type hypersensitivity responses [270].

Other strategies investigated to precondition MSCs include the use of the iron chelator deferoxamine (resulting in an increase in angiogenic factors and anti-inflammatory cytokines) [271]; antioxidant pre-treatment (N-acetylcysteine and ascorbic acid 2-phosphate, which decrease TNF-α and improve IL-10 secretion) [272]; PDGF pre-treatment (improvement in proliferation and migration) [273]; and co-culturing of MSCs and human umbilical cord extracts (Wharton’s jelly extract supernatant, which represents a cocktail of growth factors, including IGF-1, EGF, PDGF-AB, and b-FGF), L-glutamate, components of extracellular matrixes (hyaluronic acid, collagen and MUC-1) and exosomes, which contribute to preserving MSC properties and functionality [274].

Moreover, MSCs can also be tested as carriers of genes or genetic modifications. Due to their ability to migrate to injury sites, MSCs represent a promising strategy for the delivery of genes associated with the regeneration and repair of renal tissue, functioning as a “Trojan horse” [275]. Thus, several genes associated with trophic factors may be used for these purposes, such as Klotho and other trophic factors with renoprotective potential.

MSC-based therapy is equally promising in the context of genetic modifications for treating kidney diseases, including the overexpression of sirtuin-3 [276], erythropoietin, C-X-C chemokine receptor 4 (CXCR4), CTLA4Ig and IL-10/selectin, as well as the transfection of minicircles containing biological drugs, such as etanercept, which is a TNF-α blocker [277], and the transfection of nanoparticles containing iron oxide, polymers and plasmids [278].

Stromal-derived factor-1 (SDF-1), also known as CXCL12, and its receptor, the CXCR4 axis, is a crucial key pathway in cell trafficking during kidney damage. Thus, in acute kidney injury, SDF-1 mRNA levels increase more than 2.5-fold and remain as high as ~2.0-fold after 24 h within kidney cortex tissue [279]. That increase leads to homing and migration of CXCR4-expressing MSCs to the injured kidneys. However, MSCs, which express CXCR4, migrate to damaged tissues with limited efficiency. Therefore, CXCR4 gene-modified BM-MSCs lead to accumulation of these cells within the injured kidney and activation of the PI3K/AKT and MAPK signaling pathways through phosphorylation [280]. Likewise, exosomes secreted from CXCR4-overexpressing MSCs promoted an increase in angiogenesis, as opposed to a decrease in apoptosis and fibrosis [281]. These approaches represent a promising strategy for advancing MSC-based therapy.

Although MSC-based preconditioning treatment has not been associated with harmful effects, further studies are required to verify its effectiveness in maintaining MSC properties.

Looking forward, the production of a large amount of MSCs is a challenge in the advancement of MSC-based therapy in renal diseases, as multiple infusions might be required, in particular for chronic kidney disease. Therefore, automated hollow-fiber bioreactors and microcarriers have been validated for the development of the large-scale manufacturing of MSCs, providing cells with preserved characteristics and functionality in a 3D environment when compared to the manual multilayer flask method [282,283]. Importantly, this approach may be cost- and time-saving. In addition, a scaffold structure could influence the functions of seeded cells. When a hydrogel, a porous scaffold of the alginate solution (derived from brown algae) and a tissue culture plastic surface were compared, it was observed that the porous scaffold allowed the highest concentration of cytokines and growth factors produced by MSCs [284].

### 4.2. Crosstalk between Klotho and Mesenchymal Stem Cells

In addition to the previously discussed strategies designed to increase Klotho levels, MSCs are a promising strategy aiming to modulate Klotho expression in CKD and AKI [285]. Moreover, their secretome is composed of molecules such as growth factors and cytokines, which confer immune modulation and antifibrotic roles [256,286].

A preclinical study, for instance, has been conducted with streptozotocin-induced DKD in rats, which were further injected with adipose-tissue-derived mesenchymal stem cells (AT-MSCs). It was found that these cells resulted in a reduction of typical histological alterations in the animals treated, such as tubulointerstitial fibrosis and glomerular hypertrophy. Furthermore, the authors observed a decrease in apoptosis in renal tissue, as seen by a restoration of Bax and Bcl-2 to normal levels. Interestingly, the researchers also evaluated the levels of Klotho after the intervention proposed and reported that in AT-MSC animals, the levels of this protein were higher, whereas the expression of proteins such as Wnt1 and active β-catenin were lower. A similar result has been observed in vitro, in experiments involving the co-culturing of NRK52-E cells—rat kidney clones from epithelioid and fibroblastic cells—and AT-MSCs in high-glucose medium. In these in vitro experiments, a higher expression of Klotho was reported, alongside the inhibition of upregulated proteins by the hyper-glucose medium, such as Wnt 1 and active β-catenin, when the stem cells were used. In brief, it has been demonstrated that stem cells are able to mitigate renal damage through both the upregulation of Klotho and the inhibition of Wnt/-catenin signaling [266].

Concerning the inhibition of the Wnt/β-catenin pathway, it has been proposed by other studies that Klotho is able to bind to ligands of this pathway, along with the fact that it is negatively related to Klotho expression [175,182]. In an in vitro study, for instance, it has been reported that through binding, Klotho inhibited Wnt activities and consequently β-catenin and the expression of its target genes. On the other hand, the group also observed that in tissues derived from Klotho-deficient animals, Wnt signaling was higher and this resulted in cell senescence [182]. Thus, considering Klotho as an antagonist for the Wnt pathway, as well as the previously discussed data, it can be proposed that the downregulation of this protein in CKD might contribute to the excessive activation of Wnt/β-catenin signaling, which is associated with the development of renal conditions, such as DKD.

In addition to the reports mentioned, studies have shown that Klotho improves BM-MSC proliferation [287,288,289] and pluripotency potential by increasing the expression of pluripotent transcription factors such as OCT4 and Nanog [289] and reduces their osteogenic potential in vitro through the downregulation of FGFR1 and decreased phosphorylation of ERK1/2 [288]. Moreover, Ullah, et al., observed higher senescence and apoptosis rates, related to Klotho deficiency in AT-MSCs. Decreased Klotho expression is associated with the impairment of telomerase activity and the upregulation of pro-inflammatory and tissue-remodeling-related gene expression and SRY box-2 (Sox2) and CD90 [290]. Therefore, restoring Klotho expression may preserve MSC function and enhance its therapeutic potential.

Furthermore, it has been shown that modification through overexpression of the Klotho gene increases the paracrine ability of BM-MSCs, expressing higher levels of IGF-1, VEGF and HGF [287]. Thus, the synergism between Klotho and MSCs may enhance the properties of each other, decreasing oxidative stress and creating a proregenerative microenvironment. In fact, the administration of modified MSCs expressing Klotho lead to higher renal superoxide dismutase (SOD) and reduced malondialdehyde (MDA) expression [291] and improved antifibrotic capacity in IRI through inhibition of the Wnt/β-catenin pathway and EMT of the epithelial tubular cells. Likewise, in an AKI mouse model of rhabdomyolysis, the modified MSCs with the Klotho gene presented markedly higher therapeutic potential when compared to the transplantation of MSC-EVs and MSCs alone. The treatment significantly reduced serum creatinine and BUN levels and preserved renal function [292].

In short, the increases in Klotho levels in kidney diseases have been associated, in different studies, with the improvement of histological damage, such as glomerular hypertrophy and tubulointerstitial fibrosis, along with reduced apoptosis and activation of the Wnt/β-catenin pathway, which is involved in the progression of kidney illnesses. At the same time, data in the literature point out that stem cells are able to modulate processes related to the attenuation of renal injury in kidney diseases, including through the regulation of Klotho expression, through their characteristic properties [266], although the improvement of their therapeutic efficacy in vivo is challenging and still requires the development of strategies, as reviewed by Yun C. W et al. [293]. When genetically modified, MSCs can be also used as a vehicle to promote the expression of Klotho. Thus, combining MSCs and Klotho represents a relevant therapeutic strategy for the treatment of kidney illnesses.

### 4.3. Perspectives on MSC and Gene Therapy for Chronic and Acute Kidney Disease

As previously discussed, CKD is considered a public health problem, with a rising incidence, that usually results in ESKD and other complications. There is a paucity of biomarkers that can be used for the early diagnosis of this condition. Therapeutic approaches are also limited and CKD usually progresses in patients. Therefore, considering that CKD is a complex disease, the development of new strategies for the treatment of this illness is of pivotal importance and the use of more than one therapeutic measure might be interesting. Similarly, AKI might lead to a higher risk of CKD and ESKD development and it is associated with high mortality in patients as well, both in hospitalized individuals and in those who are not hospitalized. Its incidence is also rising; however, there is a lack of therapies available and a lack of biomarkers for the early stages of the disease. Thus, the study of new potential therapeutic approaches for both diseases remains relevant.

In this regard, MSC therapy is an emerging approach for the treatment of acute and chronic kidney diseases. Due to their properties, these cells are an object of interest for many ongoing studies and clinical trials [294,295]. Indeed, the number of registered clinical trials using cell therapy and submissions to the US Food and Drug Administration (FDA) has recently increased for several different pathologies.

#### 4.3.1. MSC-Derived Extracellular Vesicles

As previously explored in this review, an important point of interest concerning MSC therapy is the administration of MSC-derived EVs, since most of these cells’ pro-regenerative effects occur because of their paracrine signaling. Hence, this is a promising approach for the treatment of both CKD and AKI. In addition to the interesting properties mentioned above in relation to their microRNA cargo, these vesicles exhibit a preferential tropism to the damaged kidney. Studies have demonstrated, for example, that in comparison with sham controls, in which the vesicles accumulate mainly in the liver, the damaged tissue uptakes the MSC-EVs more efficiently [296,297]. Furthermore, the treatment mitigates AKI in a dose-dependent manner through the reduction of serum creatinine levels, interstitial infiltrate and apoptosis, as well as through the promotion of tubular cell proliferation and antioxidant effects [296,298].

Similarly, it has been found that the administration of MSC-EVs from human umbilical cord-derived MSCs into a mouse model of IRI reduces cell death and induces the proliferation of tubular cells [296]. Characterization of the EVs revealed the high expression of miR-125b-5p, which promoted repair through the inhibition of tumor protein suppression p53 signaling, and thereby mitigated G2/M arrest and limited apoptosis, as will be addressed in the next topic of this review.

Furthermore, data in the literature have shown that the downregulation of microRNA in MSC-EVs decreases the protective properties of these cells in AKI. MSC-EVs containing microRNA in its inactive form failed to promote healing in a murine glycerol model of kidney injury, whereas EVs carrying mature and active microRNA ameliorated the damage, reduced the apoptosis of epithelial tubular cells and promoted a proregenerative expression profile in the injured tissue [299].

Interestingly, a review of studies with DKD murine and primate models showed an increase in survival and an improvement of functional and structural parameters after MSC transplantation [256] MSC- and MSC-derived EV-based therapies have the potential to decrease fasting blood glucose and glycated hemoglobin, thus reducing glucotoxicity and mesangial expansion. Additionally, MSC attenuates the oxidative, pro-inflammatory and remodeling milieu through the reduction of IL-6, MCP-1, IL-1β, TNF-α, TGF-β and collagen I/IV expression [300,301], indicating that these cells possess broad beneficial effects in the kidney and pancreas in DM setting.

Still concerning DKD patients, MSC-based clinical trials reported significant improvements in renal function, with decreased levels of serum creatinine, blood urea nitrogen (BUN), albuminuria and kidney hypertrophy [300]. Moreover, the treatment showed the sustained reduction of blood glucose levels and insulin requirements. According to some studies, this occurs because MSCs and MSC-derived EVs reach the injured pancreas and restore glucose homeostasis through the preservation of β-cell function and the promotion of its proliferation [302,303].

Hence, the administration of MSCs-EVs is a potential approach to MSC therapy for kidney diseases. As shown in several studies discussed here and some others that will be discussed in the next section, these vesicles present tropism for injured tissue, alongside the beneficial effects provided by their cargo.

#### 4.3.2. MicroRNAs

Regarding the previously discussed information, the MSC-derived secretome functions as a mechanism of cell–cell communication by which MSCs can modulate tissue repair and regeneration. This secretome is composed of a wide range of soluble factors and vesicles carrying cytokines, chemokines, growth factors and proteases, as well as messenger RNA (mRNA), microRNA(miR) and DNA [256]. Among the factors mentioned, the extracellular RNA carried by EVs, composed of mRNA and miRNA—a class of small non-coding RNAs that modulate gene expression and post-transcriptional modifications [304]—is a special point of interest [299,305].

Some examples of abundant microRNAs in MSC-derived EVs include miR-125a-5p, miR-26a-5p and miR-191-5p, which may have an important role in renal physiology and repair. Among these molecules, miR-125a-5p acts on the attenuation of the inflammatory response as it promotes the polarization of macrophages M2 by targeting tumor necrosis factor receptor (TRAF) 6 mRNA and regulating its expression [306,307]. Thereby, this microRNA is negatively regulated in conditions related to acute inflammation, such as LPS-induced kidney injury [307].

Moreover, miR-26a-5p mediates the maintenance and homeostasis of glomerular cell types [308]. It is highly expressed in podocytes and it contributes to their physiology by preventing extracellular matrix accumulation through the inhibition of connective tissue growth factor (CTGF), which is induced by TGF-β in a Smad3-/Smad4-dependent manner [309,310]. However, miR-26a-5p expression decreases in injury related to podocyte injury, for instance in DKD and glomerulonephritis [308]. In contrast, a high glucose milieu increases miR-26a-5p expression in mesangial cells, in which the microRNA promotes hypertrophy and extracellular matrix expression by targeting the phosphatase and tensin homolog (PTEN)/protein kinase B (Akt)/mammalian target of the rapamycin complex (mTORC)1 pathway [311]. However, renal injury in diabetes is also characterized by apoptosis of mesangial cells and, therefore, the miR-26a-5p increase in these cells may act as a compensatory mechanism to mitigate the damage. The molecule miR-26a-5p may also exert a beneficial role through the modulation of the inflammatory response. It suppresses IL-6 expression and induces regulatory T cells (Tregs) expansion in AKI and CKD [308]. In addition, miR-191-5p may act as an upholder of kidney function and homeostasis against AKI and CKD as well. It has been observed that the administration of an miR-191-5p mimic in an AKI model caused diminished cytokine expression and lowered the apoptosis rate by targeting oxidative stress responsive 1 (OXSR1) [312]. Furthermore, miR-191-5p expression is downregulated in CKD patients and it is positively correlated with eGFR [313].

Thus, as indicated by several studies, some miRs might be unbalanced in renal diseases. On the other hand, they are able to promote the amelioration of the inflammatory burden in the renal tissue, as well as the prevention of extracellular matrix accumulation. They are also important for the maintenance of cells’ physiology, such as that of podocytes. Therefore, among the components of MSCs-EVs, miRs present important roles in kidney repair.

#### 4.3.3. MSCs Combined with Sodium-Glucose Co-Transporter-2 Inhibitors

In conjunction with the above information, as further evidence of its emergent potential as a complementary strategy to the current treatment, a study has shown decreased sodium-glucose co-transporter-2 (SGLT2) expression on renal tubular cells after MSC administration in rhesus monkeys [314]. SGLT2 is the main molecule responsible for glucose reabsorption and it is a target of SGLT2 inhibitors. In fact, MSC effects were comparable to SGLT2 drug inhibition both in vivo and in vitro. Therefore, combining MSC-based cell therapy with SGLT2 inhibitors is a cutting-edge approach, as these drugs halt DKD progression and the occurrence of cardiovascular events [315]. Importantly, emerging data indicate that SGLT2 inhibitors also curtail the progression of other causes of CKDs, such as hypertension and glomerular diseases [316]. Furthermore, a meta-analysis has shown that these drugs are able to avoid AKI in patients with type 2 diabetes mellitus [317], which highlights the importance of SGLT2 inhibitors in clinical practice.

#### 4.3.4. Klotho, microRNAs and Genetically Modified MSCs in Chronic and Acute Kidney Disease

In addition to the topics discussed in this review, Klotho expression is also important in the setting of CKD and AKI. As addressed here, Klotho overexpression has renoprotective effects, whereas its downregulation is associated with the progression of kidney diseases. Interestingly, individuals with kidney injury have dysregulated miRNA expression and activity that may impair Klotho expression through epigenetic and post-transcriptional mechanisms. In fact, it has been found that diabetic patients exhibit higher levels of miR-200c, which is related to endothelial dysfunction and Klotho downregulation under oxidative stress conditions [318,319]. In addition, many miRNAs that support damage by decreasing Klotho expression have been discovered, such as miR-34a [320], miR-339 [321], miR-556 [321], miR-199b-5p [322] and miR-199a [292]. Therefore, the expression of many protective microRNAs is unbalanced in both acute and chronic conditions.

Accordingly, the transplantation of MSCs genetically modified with the Klotho gene would be a strategy to restore and promote tissue repair through different mechanisms and signaling pathways. Again, MSCs can act as therapeutic carriers for gene delivery and genetic engineering products, functioning as a “Trojan horse” [295]. Moreover, as MSCs do not express Klotho [237,289,291] it would be possible to combine Klotho’s antioxidant effects through gene therapy and the protective properties of MSCs, representing a strategy to both treat kidney diseases and preserve the therapeutic potential of MSCs.

#### 4.3.5. Challenges for the Clinical Application of MSCs

Despite the promising results in relation to clinical applications, MSC therapy still has some challenges to overcome. The heterogeneity of pre-clinical and clinical studies and features of cell sources may hamper the establishment of a standardized protocol for the procedure [256,257,295]. The potential of autologous, allogeneic, syngeneic and xenogeneic MSCs in DKD models, for instance, has been explored in reviews and positive results have been found [256]. However, the cell source is of great concern as the patient’s condition may affect the therapeutic performance of the cell, especially in patients with metabolic dysfunction, such as diabetes. MSCs isolated from these individuals present higher rates of senescence and apoptosis, decreased proliferation and angiogenesis potential, as wel as a pro-inflammatory phenotype and an altered miRNA expression profile [256,323,324]. In addition, metabolic conditions and aging can also compromise MSCs’ mRNA and miRNA expression profiles. Studies have demonstrated, for example, that metabolic syndrome increases the adipogenic differentiation and senescence of AT-MSCs, which is correlated with increased gene expression linked to apoptosis, inflammation mediated by chemokine and cytokine signaling and ubiquitin-proteasome pathways [325]. Complementarily, the impaired therapeutic potential of MSC-EVs derived from an old animal model of CKD has also been demonstrated [324]. The EVs had reduced antifibrogenic capacity associated with lower levels of miR-294 and miR-133, which attenuates TGF-β1-mediated epithelial-to-mesenchymal transition (EMT) through the suppression of Smad2/3 and ERK1/2 phosphorylation.

Lastly, in order to continue advancing MSC-based therapy, the production of a large amount of these cells is a key aspect. MSCs cultured for a prolonged period may be affected by a disturbance in their cellular structure and function. Thus, chromosomal instability and aberrations of MSCs are controversial, as some researchers have documented their occurrence [326] and others have not noted these changes [327,328]. Importantly, chromosomal evaluation is a turning point and, if present, this should lead to cell disposal. The malignant transformation of MSCs has not been described in clinical trials [329,330], even when the follow-up was carried out from 30 days to 6.8 years [249,331,332,333,334] and MSC-based cell therapy was performed for acute and chronic diseases [246,329].

## 5. Conclusions

The restoration of Klotho levels is a strategy designed to ameliorate the damaging renal microenvironment and gene expression profile present in both chronic and acute kidney diseases. Likewise, MSCs present great potential as allies for this therapeutic approach, due to their characteristic properties—including the release of EVs—and also because of their possibility to act as gene carriers. There are some challenges to be overcome in regard to the use of these cells in vivo, such as the influence of their potency on their biological performance, as well as their large-scale production and safety issues. To date, however, no clinical trials have reported malignant transformations of these cells. Therefore, the synergism between Klotho and MSCs might be an interesting therapeutic strategy for reducing the burden of CKD and AKI, in particular when associated with the current pharmacological and non-pharmacological approaches.

## Figures and Tables

**Figure 1 pharmaceutics-14-00011-f001:**
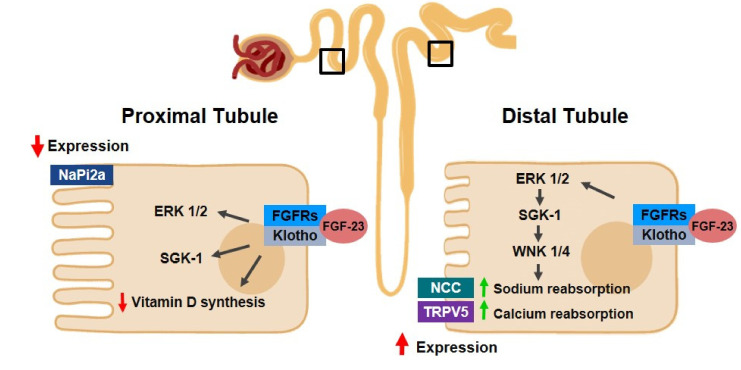
Klotho and FGF23 in mineral homeostasis in kidneys. In proximal tubules, the Klotho/fibroblast growth-factor (FGF) 23/fibroblast growth factor receptors (FGFRs) complex activates extracellular signal-regulated kinase (ERK) 1/2, serum/glucocorticoid-regulated kinase (SGK)-1 and with no lysine kinase (WNK) 1/4 pathways, which results in the reduction of the expression of sodium phosphate co-transporter (NaPi2a), leading to phosphaturia. In distal tubules, in turn, the same complex and signaling pathways are activated and this results in an increase in sodium chloride cotransporter (NCC) and transient receptor potential cation channel subfamily V member 5 (TRPV5) channels, which contributes to increases in both sodium and calcium reabsorption, respectively.

**Figure 2 pharmaceutics-14-00011-f002:**
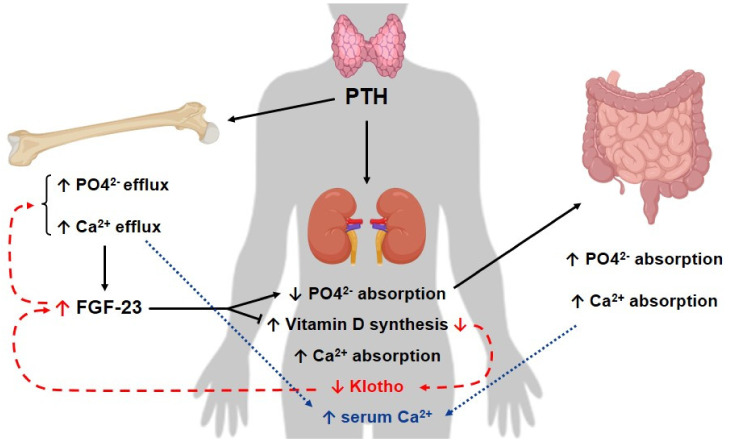
Klotho/FGF and PTH axis in CKD. In kidneys, parathyroid hormone (PTH) leads to an increase in calcium (Ca^2^+) absorption and Vitamin D synthesis, whereas it diminishes phosphorus absorption. On the other hand, PTH stimulates both phosphorus and calcium efflux in bones. In turn, PTH-stimulated Vitamin D production increases the gastrointestinal reabsorption of these minerals. As a result, both gastrointestinal calcium reabsorption and its efflux from bones contribute to a rise in calcium excretion. Phosphorus (PO_4_^2^^−^) is also eliminated as a consequence. In chronic kidney disease (CKD) (red dotted line), there is a reduction in Klotho expression, alongside a decrease in Vitamin D levels and an increase in fibroblast growth factor (FGF-)23 levels. It is important to mention that the decrease of Vitamin D is related to the decrease of Klotho in the kidneys, which then leads to a rise in FGF-23 levels in the bones. As a consequence, this hormone diminishes Vitamin D production. As a result of this axis dysregulation, the inhibition of PTH synthesis promoted by these components is lost, which leads to a rise in the levels of this hormone, which also contributes to the elevation of these molecules. Secondary hyperparathyroidism associated with CKD might be a result of the described imbalance in this axis for CKD patients.

**Figure 3 pharmaceutics-14-00011-f003:**
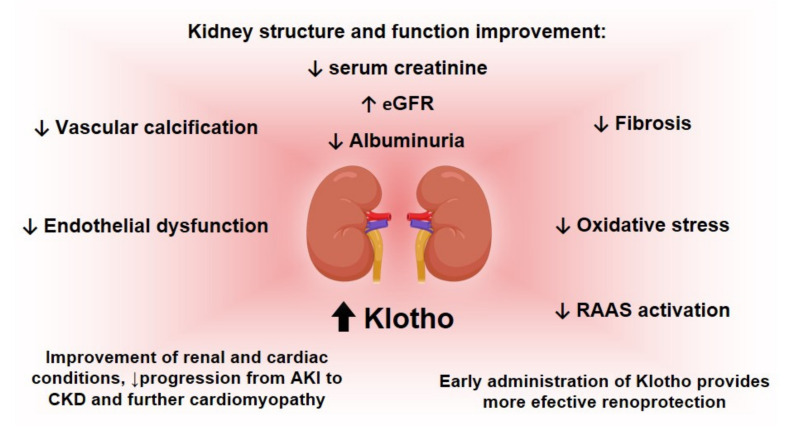
Renal benefits after restoration of Klotho levels. The figure shows some of the benefits for kidneys obtained with the reestablishment of Klotho levels, concerning the progression of acute kidney disease (AKI) and chronic kidney disease (CKD) and other phenotypes associated with these diseases. Klotho ameliorates processes involved in the advancement of renal diseases, such as the inhibition of the renin-aldosterone-angiotensin system (RAAS), fibrosis and oxidative stress. In addition, Klotho also reduces vascular calcification and vascular dysfunction and it promotes the improvement of renal structural and functional conditions, such as an increase in the estimated glomerular filtration rate (eGFR) and a decrease in serum creatinine.

**Figure 4 pharmaceutics-14-00011-f004:**
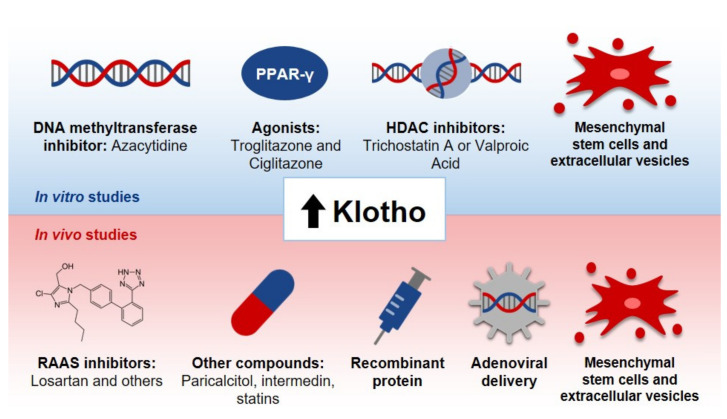
Possible therapeutic approaches for the restoration of Klotho levels. The figure summarizes some of the potential strategies to increase Klotho levels, considering both endogenous and exogenous alternatives for that purpose, as reviewed by Buchanan, et al. [2].

**Figure 5 pharmaceutics-14-00011-f005:**
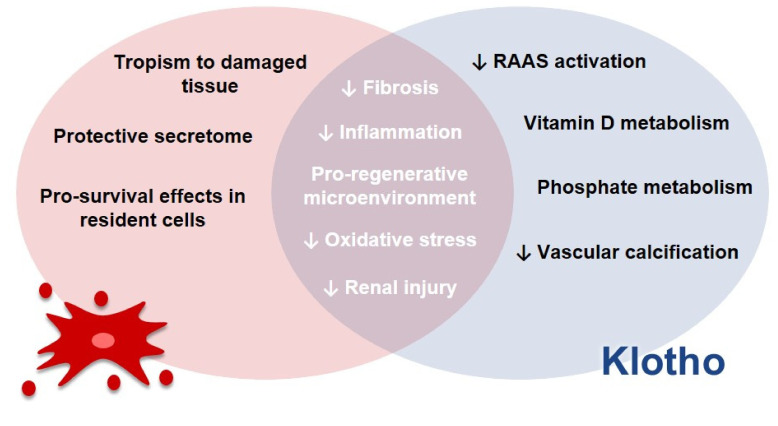
Synergism between mesenchymal stem cells (MSCs) and Klotho. We speculate, based on data in the literature, that, in vivo, the properties of MSCs, such as tropism to damaged tissues, their protective secretome and pro-survival effects in resident cells, would act alongside the beneficial effects of Klotho, such as its reduction of renin-angiotensin-aldosterone-system (RAAS) activation, its influence on vitamin D and phosphate metabolism and the reduction in vascular calcification, As a result, there would be reduction in fibrosis and inflammation, improvement of pro-regenerative conditions for the damaged tissue, as well as a reduction in oxidative stress and renal injury in general.

**Table 1 pharmaceutics-14-00011-t001:** Klotho and the Klotho/FGF-23 axis in CKD-associated cardiovascular disease. Summary of some of the studies analyzed in this review, concerning the role of Klotho and the Klotho/FGF-23 axis in cardiovascular disease, an important and common morbidity and cause of mortality in CKD patients.

Author/Year	Model Used	Study Design	Conclusion
Karalliedde, J., et al., 2013 [85]	Patients with diabetes type 2, presenting systolic hypertension and albuminuria	Single-Center, Double-Blind Randomized Controlled Trial	Inhibition of RAAS led to an increase in soluble Klotho levels.
Saito, Y., et al., 2000 [86]	Rats with atherosclerosis	Preclinical Study	Klotho adenoviral delivery resulted in the mitigation of vascular endothelial dysfunction and reduction of blood pressure values
Kuro-o, M., et al.,1997 [1]	Klotho-deficient mice	Preclinical Study	The animals presented artery calcification, cardiac fibrosis and hypertrophy. Klotho might participate in the signaling pathways involved in these processes.
Xie, J., et al., 2015 [87]	Klotho-deficient mice	Preclinical Study	The increase in soluble Klotho levels attenuated cardiac remodeling in CKD animals. Decrease in this protein level is proposed to be an independent factor for cardiomyopathy in CKD.
Ding, et al., 2019 [88]	Mice with angiotensin-II infusion	Preclinical Study	Klotho was related to the decrease of cardiac FGF-23 expression in vitro and in vivo; moreover, it prevented cardiac remodeling and dysfunction in this model.
Memmos, et al., 2019 [84]	79 patients on dialysis	Prospective Cohort Study	Low levels of Klotho are correlated with an increased risk of cardiovascular disease and reduced overall survival in these patients. It might contribute to cardiovascular disease in individuals with CKD.
Brandenburg, V.M., et al., 2015 [89]	2948 patients	Multicenter Longitudinal Study	In individuals with normal kidney function, Klotho does not act as a predictive marker of cardiovascular and mortality risk.
Pan, H.C., et al., 2018 [90]	168 patients with diabetes type 2	Prospective Study	Low levels of Klotho are associated with cardiovascular outcomes, such as coronary disease. In these patients, Klotho level is a predictor for vascular events.
Gutierrez, O.M., et al., 2009 [91]	162 patients with CKD	Cross-Sectional Study	FGF-23 is correlated with vascular dysfunction, such as left ventricular mass index and hypertrophy in these individuals.

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
