# Peer review of "Klotho and Mesenchymal Stem Cells: A Review on Cell and Gene Therapy for Chronic Kidney Disease and Acute Kidney Disease"

_pharmaceutics, 2021, doi:10.3390/pharmaceutics14010011_

Round 1
Reviewer 1 Report
In this review, the role of Klotho in the chronic kidney disease (CKD) and acute kidney disease (AKD) is explained. Some researches targeted to Klotho are introduced as the promising therapy. In addition, it is introduced that the cell therapy using mesenchymal stem cells (MSCs) is effective to restore the expression level of Klotho. The authors explain the details of relationship between Klotho and CKD and AKD. However, the information of MSC therapy is insufficient. Introduction on the MSC therapy is entirely poor. Some appropriate references should be quoted while the Klotho merit should be clearly explained by comparing with the corresponding therapeutic strategies. Taken together, major revision should be made before the paper re-submission.
1. In the section of 2.2, it is claimed that the approach controlled the expression level of Klotho is “safe”. How safe it is? What is the evidence? The scientific evidence should be shown.
2. In the chapter 4, by quoting [2], it is introduced that there are various methods for regulating the expression level of Klotho. What are the advantages or disadvantages of these methods? Why is the MSCs’ treatment focused on among various methods?
3. In the chapter 5, it should be explained that the detail of the mechanism in which the MSC promote the expression of Klotho and inhibit the Wnt/-catenin signaling.
4. In the chapter 6, when applying MSCs’ treatment to renal diseases, what is needed to increase the therapeutic effect? Reference [210] introduces only the general problems of MSC therapies. Identified the renal, what is the required technology?
5. In the chapter 7, some sentences about the details of references [219-222] should be added. It is interesting for readers to know how the main five miRNA contained EVs derived MSCs work in the improvement of AKI.
6. In the chapter 7, it is mentioned that the MSC therapy is a safe procedure. What is the scientific evidence to say “safe”? Appropriate references should be quoted.
7. In the chapter 7, the mechanism in which the over-expression of Klotho improves MSCs’ functions should be described. What is the advantage of Klotho treatment maintain and improve the therapeutic functions of MSCs? The advantage or disadvantage of this treatment to compare with other strategies should be described clearly.
8. Regarding the overall composition, I think the chapters 4, 5, 6, and 7 should be combined into one chapter to describe the MSC and gene therapy for CKD and AKD.
9. There is one typographical errors. On the 1st line in Table 1, the word “na” should be “Na”.
Author Response
We would like to thank the reviewers for their comments, as they contribute to strengthen our manuscript. We acknowledge the concerns raised in the review and have carefully addressed the reviewers’ comments and suggestions, as (a) reorganization of the structure of the manuscript and figures; and (b) addition of new paragraphs and references regarding the efficacy and safety of MSCs and Klotho.
All changes were performed through TrackChanges platform.
Reviewer 1
In this review, the role of Klotho in the chronic kidney disease (CKD) and acute kidney disease (AKD) is explained. Some researches targeted to Klotho are introduced as the promising therapy. In addition, it is introduced that the cell therapy using mesenchymal stem cells (MSCs) is effective to restore the expression level of Klotho. The authors explain the details of relationship between Klotho and CKD and AKD. However, the information of MSC therapy is insufficient. Introduction on the MSC therapy is entirely poor. Some appropriate references should be quoted while the Klotho merit should be clearly explained by comparing with the corresponding therapeutic strategies. Taken together, major revision should be made before the paper re-submission.
- In the section of 2.2, it is claimed that the approach controlled the expression level of Klotho is “safe”. How safe it is? What is the evidence? The scientific evidence should be shown.
Response: Thank you for your suggestion. We addressed the safety aspects of Klotho in the 2nd and 3rd paragraphs of page 4.
- In the chapter 4, by quoting [2], it is introduced that there are various methods for regulating the expression level of Klotho. What are the advantages or disadvantages of these methods? Why is the MSCs’ treatment focused on among various methods?
Response: Thank you for bringing out this topic to discussion. We expanded the topic on pages 23-24. Likewise, please verify the topics about MSC safety (page 35), efficacy/potency (pages 27-28 and 30-31) in preclinical studies and clinical trials, which indicate that these cells are a promising strategy not only in kidney diseases but also in other acute and chronic kidney diseases.
- In the chapter 5, it should be explained that the detail of the mechanism in which the MSC promote the expression of Klotho and inhibit the Wnt/-catenin signaling.
Response: As suggested by the reviewer, we added some sentences explaining the interactions between Klotho and Wnt/catenin signalling in the second paragraph of page 32.
- In the chapter 6, when applying MSCs’ treatment to renal diseases, what is needed to increase the therapeutic effect? Reference [210] introduces only the general problems of MSC therapies. Identified the renal, what is the required technology?
Response: Thank you very much for your suggestion. We added new information on pages 30-31.
- In the chapter 7, some sentences about the details of references [219-222] should be added. It is interesting for readers to know how the main five miRNA contained EVs derived MSCs work in the improvement of AKI.
Response: Thank you for bringing this topic to discussion. We expanded the explanation about the main miRNA overexpressed in that reference and we also included other references that address other miRNAs in AKI on pages 33-34.
- In the chapter 7, it is mentioned that the MSC therapy is a safe procedure. What is the scientific evidence to say “safe”? Appropriate references should be quoted.
Response: Thank you very much for your suggestion. We added new information about the safety of MSCs on page 35, as well as the appropriate references.
- In the chapter 7, the mechanism in which the over-expression of Klotho improves MSCs’ functions should be described. What is the advantage of Klotho treatment maintain and improve the therapeutic functions of MSCs? The advantage or disadvantage of this treatment to compare with other strategies should be described clearly.
Response: We would like to thank the reviewer for the important comments. We ask you to, please, verify our comments previously on pages 23-24 and 27-31 (section 4). Also, in addition to the previous comment, we added the following sentences in the section regarding MSCs, on page 34:
‘’ MSCs can act as therapeutic carriers for gene delivery and genetic engineering products, functioning as a “Trojan Horse” [221]. Moreover, MSCs do not express Klotho (Zhang et al, 2018; Xie et al, 2019; Grange et al, 2020). Therefore, combining Klotho antioxidant effects through gene therapy and the protective properties of MSC is a strategy to both treat kidney diseases and preserve MSC therapeutic potential’’.
Likewise, we added a new figure (Figure 5) describing the potential synergism between Klotho and MSCs.
- Regarding the overall composition, I think the chapters 4, 5, 6, and 7 should be combined into one chapter to describe the MSC and gene therapy for CKD and AKD.
Response: We would like to thank you the reviewer for this suggestion. We combined the chapters 4 to 7 in chapter 4 entitled “Therapeutic Potential of Klotho in Acute and Chronic Kidney Diseases”.
- There is one typographical errors. On the 1stline in Table 1, the word “na” should be “Na”.
Response: Thank you. The typo was corrected.
Reviewer 2 Report
The review is interesting; a lot of information is presented. However, the review needs better structuration and summarization of main findings or presented data. Some chapters and figures should be interchanged their places. Review contains many factual data, therefore each paragraph needs finalization of findings, stressing the main mechanisms or the Klotho effects. I would suggest to separate MSC part for another article.
Minor remarks:
- The manuscript title does not fit to its content – the MSCs and gene therapy is a small part of this review and the last chapter, while the whole review was about the chronic and acute kidneys diseases, patients and the effects of Klotho. The part of the review about the MSC is superficially written.
- Since everything is about renal diseases, the paragraph „2.2.3. Cardiovascular Disease and CKD“ would be more appropriate to change into the “CKD and cardiovascular diseases”
- The “2.4. Klotho, NF-κB and TGF-β“ paragraph would be more informative if the authors would separate into: for example: „Klotho and inflammation“ and „Klotho and fibrosis“. The inflammation part needs more data.
- The paragraph 2.2.5. Klotho and Angiotensin II can be a part of a paragraph „CKD and cardiovascular diseases”.
- Each paragraph should be more concrete and clear summarized. It is not clear now, what the main mechanisms or findings are.
- If there are the non-inflammatory and inflammatory mechanisms of AKI, the review should be structurally separated into these two types. What the mechanisms are typical for the non-inflammatory AKI and how Klotho affects them, the clear stating is missing.
- Page 14, the Klotho needs capital letter.
- All abbreviations used in Figures should be extended in legends.
- Figures 3 and 4 should be interchanged their places and better described in the text. The chapter “Restoration of Klotho Levels: a Potential Therapeutic Approach“ is quite weak and should be separated in two parts: the restoration and therapeutic approach.
- Figure 3 probably should include the administration trough the MSC part. Why only this delivery pathway is mentioned? Why the authors are not talking more detailed about other Klotho deliver pathways?
- First the authors are talking about the “ Klotho Administration through Mesenchymal Stem Cells“ and only then explain what the MSCs are.
- The authors are talking only about Klotho delivery by Adipose-Derived Mesenchymal Stem Cells, while more types of MSCs are known.
- The MSC parts is also very week and not informative (superficially presented data). Seems that clinicians, not cell biologists, were writing this review. I would suggest to leave the MSC part for another article.
- Why the exosomes were measured if no Klotho data were presented. It is also a deliver pathway but it is not mentioned in the Figure 3.
- It is still not clear how the Klotho can be delivered or affected trough the MSC if the MSC by themselves have anti-inflammatory and many other paracrine effects? The chapter” Perspectives on MSC and Gene Therapy for Chronic and Acute Kidney Disease“ is presenting the effects of the MSC, but not the MSC-related Klotho effects.
Author Response
We would like to thank the reviewers for their comments, as they contribute to strengthen our manuscript. We acknowledge the concerns raised in the review and have carefully addressed the reviewers’ comments and suggestions, as (a) reorganization of the structure of the manuscript and figures; and (b) addition of new paragraphs and references regarding the efficacy and safety of MSCs and Klotho.
All changes were performed through TrackChanges platform.
Reviewer 2
The review is interesting; a lot of information is presented. However, the review needs better structuration and summarization of main findings or presented data. Some chapters and figures should be interchanged their places. Review contains many factual data, therefore each paragraph needs finalization of findings, stressing the main mechanisms or the Klotho effects. I would suggest to separate MSC part for another article.
Minor remarks:
1. The manuscript title does not fit to its content – the MSCs and gene therapy is a small part of this review and the last chapter, while the whole review was about the chronic and acute kidneys diseases, patients and the effects of Klotho. The part of the review about the MSC is superficially written.
Response: We agree that the reviewer made an important point and we performed extensive review on these topics. We expanded both parts of MSCs and Klotho, as documented on pages 27-31 and 35 (MSC) and pages 12-14 and 23-24 (Klotho).
2. Since everything is about renal diseases, the paragraph „2.2.3. Cardiovascular Disease and CKD“ would be more appropriate to change into the “CKD and cardiovascular diseases”
Response: Thank you for your observation. As suggested, we changed the subtitle 2.2.3 from “Cardiovascular Disease and CKD” to “CKD and Cardiovascular Disease”.
3. The “2.4. Klotho, NF-κB and TGF-β“ paragraph would be more informative if the authors would separate into: for example: „Klotho and inflammation“ and „Klotho and fibrosis“. The inflammation part needs more data.
Response: Thank you for the suggestion. We reconsidered this section and we splited them into “Klotho and Inflammation” and “Klotho and Fibrosis” as indicated by your comment. We rewrote the first section and we also complemented the second one as it follows on pages 13 & 14.
4. The paragraph 2.2.5. Klotho and Angiotensin II can be a part of a paragraph „CKD and cardiovascular diseases”.
Response: As suggested, we transferred the paragraph 2.2.5 Klotho and Angiotensin II to the paragraph CKD and Cardiovascular Disease. Besides, we also corrected the capital letter in “Klotho” in paragraphs seven (7) and nine (9). Lastly, considering the changes made and the suggestion of comment number five (5), we rewrote the last paragraph as it follows (page 12):
‘’All in all, the information previously presented highlights the association between Klotho and cardiovascular disease in CKD – a common complication that often contributes to mortality in chronic kidney disease patients. This event is observed through different mechanisms, as mentioned above, such as the increase of Klotho resultant from RAAS inhibition and the renoprotection conferred by this protein against renal damage induced by RAAS, whilst the activation of RAAS, on the other hand, is related to the reduced levels of Klotho, for example. Besides, decreased levels or lack of Klotho are shown to be associated with vascular calcification, LVH and cardiomyopathy. FGF-23, in turn, is also related to endothelial dysfunction and LVH. Taken together, this data indicate the necessity of further research regarding Klotho and the Klotho/FGF-23 axis in CKD patients, since the results of present studies indicate the potential of these molecules as therapeutic targets to prevent mortality in these individuals, considering their involvement in the pathophysiology of a relevant complication in CKD.’’
5. Each paragraph should be more concrete and clear summarized. It is not clear now, what the main mechanisms or findings are.
Response: In accordance with reviewer’ suggestion, we reconsidered and reviewed all the paragraphs.
6. If there are the non-inflammatory and inflammatory mechanisms of AKI, the review should be structurally separated into these two types. What the mechanisms are typical for the non-inflammatory AKI and how Klotho affects them, the clear stating is missing.
Response: We appreciate your suggestion. We reconsidered the topic and we divided them into two subsections and added the information on pages 17 to 19.
7. Page 14, the Klotho needs capital letter.
Response: We corrected the capital letter of Klotho.
8. All abbreviations used in Figures should be extended in legends.
Response: We included all abbreviations used in figures in legends. Also, we corrected ‘’serum/glucocorticoid-regulated kinase” in the first paragraph of section 2.2.1 Klotho and FGF-23 and excluded the word ‘’channel’’ in the legend of Figure 1. We corrected the abbreviation ‘’SGK’’ and ‘’FGFs’’ to ‘’SGK-1” and “FGFRs”, respectively, in Figure 1. In the legend of Figure 2, we also rewrote ‘’calcium’’ without capital letter .In Figure 2, we rewrote ‘’FGF23” as “FGF-23”. In Figure 3 (former Figure 4, changed according to the suggestion of comment number 9), we corrected ‘’GFR’’ to ‘’eGFR” and added “reduction of albuminuria”, represented through an arrow.
9. Figures 3 and 4 should be interchanged their places and better described in the text. The chapter “Restoration of Klotho Levels: a Potential Therapeutic Approach“ is quite weak and should be separated in two parts: the restoration and therapeutic approach.
Response: Thank you for your suggestion. We interchanged the places of Figures 3 and 4. We also changed the order of first and second paragraphs. In the first paragraph (former paragraph 2), we partially rewrote the sentences as it follows:
‘’The reestablishment of Klotho provides benefits concerning the progression of renal disease and other events associated with it, as it has been discussed in this review and it will be briefly shown in Figure 3.’’
We expanded the topic as well, please, verify our previous comment (Question 2, Reviewer 1, section 4).
10. Figure 3 probably should include the administration trough the MSC part. Why only this delivery pathway is mentioned? Why the authors are not talking more detailed about other Klotho deliver pathways?
Response: Thank you for your suggestion. We added MSC part in Figure 4 (previous Figure 3) and we also added a new figure made by us regarding the topic, Figure 5. We expanded the other Klotho deliver pathways as well (pages 23 & 24), we kindly ask you to, please, verify our previous comment (Question 2, reviewer 1).
11. First the authors are talking about the “ Klotho Administration through Mesenchymal Stem Cells“ and only then explain what the MSCs are.
Response: As suggested, we changed the order of sections ‘’Klotho Administration through MSCs’’ and ‘’Mesenchymal Stem Cells.’’
12. The authors are talking only about Klotho delivery by Adipose-Derived Mesenchymal Stem Cells, while more types of MSCs are known.
Response: We expanded the topic on MSCs. We kindly ask you to, please, verify our previous comments on pages 27-31 and 35.
13. The MSC parts is also very week and not informative (superficially presented data). I would suggest to leave the MSC part for another article.
Response: Please verify our comments previously (Reviewer 1, response 4 & 6; Reviewer 2, response 1). We expanded the description of MSCs, including mechanisms of action, therapeutic potential and challenges (pages 27-31, 35).
14. Why the exosomes were measured if no Klotho data were presented. It is also a deliver pathway but it is not mentioned in the Figure 3.
Response: Thank you for bringing these topics to discussion. On page 27, we added novel sentences and 8 references to highlight the importance of extracellular vesicles and Klotho. We focused on the importance of MSC-derived secretome in modulating several biological processes during kidney injury, the advantages of using this strategy and the possibility of promoting genetic modification, e.g., overexpression of Klotho, for therapeutic purpose.
In addition, we added the EVs as a delivery pathway in Figure 4 (former Figure 3).
15. It is still not clear how the Klotho can be delivered or affected trough the MSC if the MSC by themselves have anti-inflammatory and many other paracrine effects? The chapter” Perspectives on MSC and Gene Therapy for Chronic and Acute Kidney Disease“ is presenting the effects of the MSC, but not the MSC-related Klotho effects.
Response: Thank you for your comment. We expanded the discussion on MSCs and Klotho, as indicated by previous comments. We kindly ask you to, please, verify our previous comments. In addition, we added a new figure (Figure 5) to report the potential synergism between MSCs and Klotho.
Reviewer 3 Report
this is a full review on a topic quite relevant, well described with some sc schemes that facilitate comprehension
Author Response
Reviewer 3
this is a full review on a topic quite relevant, well described with some sc schemes that facilitate comprehension
Response: Thank you for supporting our study.
Round 2
Reviewer 1 Report
The authors have made the corresponding modifications and quoted the additional citations according to the first review comments. I think that this paper will be accepted by pharmaceutics.
Author Response
We would like to thank the reviewer for supporting our study.
Sincerely,
Marcella L Franco
Stephany Beyerstedt
Érika B Rangel
Reviewer 2 Report
The authors corrected the manuscript and significantly improved according to the suggestions. However, the authors still need to put some efforts.
- The English language of added sentences should be carefully checked. For example, it is hard to follow the meaning of the sentence: “ Also, there are studies that indicate that in Klotho deficient animals, there is evidence of increased Wnt signaling pathway activation” (page 14); writing of “Hsp70 “ (page 15); sentence “In a nutshell, in AKI, Klotho has been associated with the inhibition of Wnt/ β-path-way, considered, then, as an anti-fibrotic molecule, Besides, this protein has been also shown to be involved with reduction of both senescence and apoptosis, alongside the im-provement of renal parameters, in different models of study involving AKI “ (page 16) and many other.
- Some parts of review have too many authors’ names cited in the text, particularly in corrected parts, that complicates understanding of the text (for example, page 12, 14, 17, 18 and other). The authors names can be mentioned if their findings are very important but not very often (for example, as in chapter “3.1.2. Klotho and non-inflammatory mechanisms in AKI” and chapter 4. Therapeutic Potential of Klotho in Acute and Chronic Kidney Diseases”).
- The part “4. Therapeutic Potential of Klotho in Acute and Chronic Kidney Diseases” could have a bold name for each paragraph to better follow the therapeutics mentioned in Figure 4.
- The chapter “4.1. Mesenchymal Stem Cells” is way too long and should be divided into parts according to the topics that authors want to stress. In this way it is difficult to understand what the authors want to stress about the MSCs. To collect all known data in the world into one chapter without stressing the meaning is not a best way to write a review.
- The same remarks can be said about the last section – it is too long and needs to be divided into part according to the main meanings.
- The summarized conclusions or total summary of the review is missing.
Author Response
We would like to thank the reviewer for the comments.
We also thank the opportunity to resubmit our study to Pharmaceutics journal.
- The English language of added sentences should be carefully checked. For example, it is hard to follow the meaning of the sentence: “ Also, there are studies that indicate that in Klotho deficient animals, there is evidence of increased Wnt signaling pathway activation” (page 14); writing of “Hsp70 “ (page 15); sentence “In a nutshell, in AKI, Klotho has been associated with the inhibition of Wnt/ β-path-way, considered, then, as an anti-fibrotic molecule, Besides, this protein has been also shown to be involved with reduction of both senescence and apoptosis, alongside the im-provement of renal parameters, in different models of study involving AKI “ (page 16) and many other.
Response: As suggested, we reviewed all sentences added. We also corrected the typos in the term ‘’sigalling’’ to ‘’signaling’’ on pages 4, 10 and 6. On page 9, we also corrected the typo ‘’improvement’’ to ‘’improvements’’. On the same page, we also added a comma after “Moreover”. On page 13, we corrected the capital letter in Klotho in section 3.1. On page 16, we corrected the size of ‘’H2O2’’ to ‘’H2O2’’.
- Some parts of review have too many authors’ names cited in the text, particularly in corrected parts, that complicates understanding of the text (for example, page 12, 14, 17, 18 and other). The authors names can be mentioned if their findings are very important but not very often (for example, as in chapter “3.1.2. Klotho and non-inflammatory mechanisms in AKI” and chapter 4. Therapeutic Potential of Klotho in Acute and Chronic Kidney Diseases”).
Response: Thank you for your suggestion. We excluded the authors’ name in the sections and pages mentioned in your comment. Also, we checked the other sections, especially the corrected parts, and we did the same procedure.
- The part “4. Therapeutic Potential of Klotho in Acute and Chronic Kidney Diseases” could have a bold name for each paragraph to better follow the therapeutics mentioned in Figure 4.
Response: Thank you for your suggestion. We wrote a bold name for each one of the paragraphs, as suggested, as it follows (page 17-19). We also corrected the typo in the legend of Figure 4 in the term ‘’sumarizes’’ to ‘’summarizes’’.
- The chapter “4.1. Mesenchymal Stem Cells” is way too long and should be divided into parts according to the topics that authors want to stress. In this way it is difficult to understand what the authors want to stress about the MSCs. To collect all known data in the world into one chapter without stressing the meaning is not a best way to write a review.
Response: We divided this chapter into subsections ‘’4.1.1 Properties and Characterization of Mesenchymal Stem Cells’’ and ‘’4.1.2 Efficacy and Safety of Mesenchymal Stem Cells’’ as it follows. We also corrected the typographical errors in terms ‘’allogenic’’ to ‘’allogeneic’’ and ‘’signalling’’ to ‘’signaling’’ in these sections, on pages 20-25.
- The same remarks can be said about the last section – it is too long and needs to be divided into part according to the main meanings.
Response: Thank you for your suggestion. We divided the last section into subsections as it follows (page 27-31).
- The summarized conclusions or total summary of the review is missing.
Response: We reviewed the Conclusion section, as suggested.
Sincerely,
Marcella L Franco
Stephany Beyerstedt
Érika B Rangel
